# Self-Supervised Contrastive Learning
# is Approximately Supervised Contrastive Learning

**Achleshwar Luthra**      **Tianbao Yang**      **Tomer Galanti**

{luthra,tianbao-yang,galanti}@tamu.edu
Department of Computer Science and Engineering
Texas A&M University

## Abstract

Despite its empirical success, the theoretical foundations of self-supervised contrastive learning (CL) are not yet fully established. In this work, we address this gap by showing that standard CL objectives implicitly approximate a supervised variant we call the negatives-only supervised contrastive loss (NSCL), which excludes same-class contrasts. We prove that the gap between the CL and NSCL losses vanishes as the number of semantic classes increases, under a bound that is both label-agnostic and architecture-independent.

We characterize the geometric structure of the global minimizers of the NSCL loss: the learned representations exhibit augmentation collapse, within-class collapse, and class centers that form a simplex equiangular tight frame. We further introduce a new bound on the few-shot error of linear-probing. This bound depends on two measures of feature variability—within-class dispersion and variation along the line between class centers. We show that directional variation dominates the bound and that the within-class dispersion's effect diminishes as the number of labeled samples increases. These properties enable CL and NSCL-trained representations to support accurate few-shot label recovery using simple linear probes.

Finally, we empirically validate our theoretical findings: the gap between CL and NSCL losses decays at a rate of $\mathcal{O}(\frac{1}{\#\text{classes}})$; the two losses are highly correlated; minimizing the CL loss implicitly brings the NSCL loss close to the value achieved by direct minimization; and the proposed few-shot error bound provides a tight estimate of probing performance in practice. The code and project page of the paper are available at [code, project page].

## 1   Introduction

Unsupervised representation learning refers to a class of algorithmic approaches designed to discover meaningful representations of complex data without relying on explicit supervision signals. The goal is to learn representations that preserve and expose semantic information, allowing them to be effectively leveraged in downstream supervised tasks. In recent years, these methods have proven to be effective in pre-training models on unlabeled data, enabling strong generalization on downstream computer vision tasks [1, 2, 3, 4, 5, 6, 7, 8, 9, 10, 11, 12, 13, 14, 15, 16] and in natural language processing [17, 18, 19, 20, 21, 22, 23, 24].

One of the most successful paradigms for unsupervised learning is self-supervised contrastive learning (CL), where models are trained to distinguish different augmented views of the same image (positives) from views of other images (negatives), typically by minimizing the InfoNCE loss [26, 2]. Notable methods in this category include SimCLR [5], MoCo [27, 6, 7], and CPC [2]. These approaches have led to representations that generalize remarkably well to downstream tasks, often competing with or

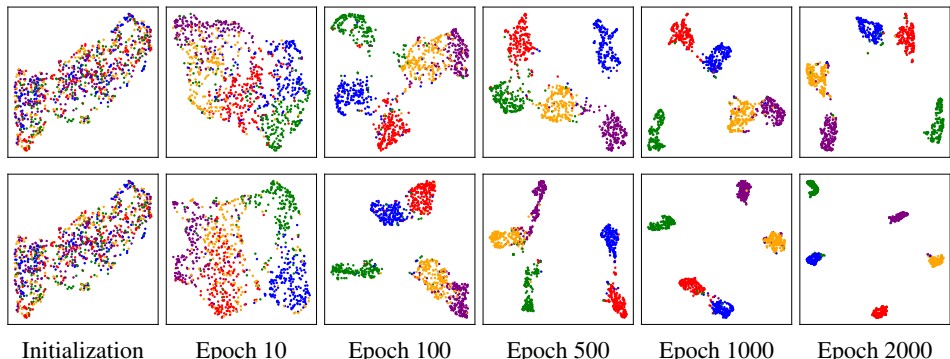

| Initialization | Epoch 10 | Epoch 100 | Epoch 500 | Epoch 1000 | Epoch 2000 |

Figure 1: *DCL forms semantic clusters without label supervision, while NSCL yields tighter, more separable clusters, despite not explicitly pulling same-class samples together.* We plot UMAP visualizations for **(top)** decoupled contrastive learning (DCL) [25] and **(bottom)** negatives-only supervised contrastive learning (NSCL) training on mini-ImageNet. See Appendix A for details.

even surpassing supervised learning. For example, an ImageNet-1M pre-trained MoCo achieves 81.5 $AP_{50}$ on PASCAL VOC [28], while its supervised counterpart achieves 81.3 $AP_{50}$.

Although the representations learned in supervised classification problems are relatively well understood–thanks to characterizations such as neural collapse [29, 30], which occurs under a variety of loss functions including cross-entropy [31, 32, 33], mean squared error (MSE) [30, 34, 35], and supervised contrastive loss [33, 36, 37, 38]–the theoretical understanding of SSL remains limited. Surprisingly, despite the absence of labels, SSL-trained models often produce representations that closely resemble those learned via supervised training: they tend to cluster nicely with respect to semantic classes [39, 40, 41] (see also Fig. 1) and support downstream tasks such as linear classification and clustering [5, 27, 6, 7, 2]. This raises the following question:

> *How does self-supervised contrastive learning learn representations similar to those learned through supervised learning, despite the lack of explicit supervision?*

**Contributions.** We provide a theoretical framework that connects CL to it supervised counterpart. Our key contributions are:

- **Duality between self-supervised and supervised CL:** In Thm. 1 we formally establish a connection between decoupled contrastive learning (DCL) [25] and a supervised variant we call negatives-only supervised contrastive loss (NSCL), in which negative pairs are drawn from different classes. Our main insight is that, although DCL does not explicitly exclude same-class samples from the loss denominator, the probability of treating them as negatives vanishes as the number of classes increases. We prove that the gap between the two losses shrinks with more classes. Unlike prior analyses [42, 43, 44, 36, 45], our result holds without assumptions on model architecture, data distribution, or augmentations. Figs. 3–4 show that the two losses are highly correlated, their gap decreases as the number of classes increases, and minimizing the DCL loss implicitly minimizes the NSCL loss.

- **Estimating the few-shot error on downstream tasks:** Prop. 1 introduces a bound on the $m$-shot classification error based on geometric properties of the learned representation. Specifically, the bound implies that lower class-distance-normalized-variance (CDNV) (see [46] and Sec. 3.2) and directional CDNV (see Sec. 3.2) lead to lower $m$-shot error, with the impact of the regular CDNV diminishing as the number $m$ of per-class labeled examples increases. Empirically, we verify this behavior for DCL-trained models (Fig. 7) and find the bound to be tight in practice (Fig. 6).

- **Characterizing the global minima of the NSCL loss:** In Thm. 2, we show that any global minimizer of the NSCL loss satisfies: **(i)** *Augmentation collapse*—all augmented views of a sample map to the same point; **(ii)** *Within-class collapse*—all samples from the same class share a representation; and **(iii)** *Simplex ETF structure*—class centers form a maximally separated, symmetric configuration. These solutions coincide with the optimal solutions of certain supervised classification settings, including those using cross-entropy loss [31, 32, 33], mean squared error

(MSE) [30, 34, 35], and supervised contrastive loss [33]. Combined with Prop. 1, these properties imply that label recovery via linear probing is guaranteed with few labels.

## 2 Related Work

**Theoretical analyses of CL.** A growing body of work aims to explain the effectiveness of CL. Early studies attributed CL's success to maximizing mutual information between augmentations of the same image [3], but later work showed that enforcing strict bounds on mutual information can hurt performance [47, 48]. Another influential line of work focuses on the alignment and uniformity of learned representations [49, 50, 51]. For example, [49] showed that same-sample augmentations are aligned, while negatives are uniformly distributed on a sphere. [51] extended this to a wider family of losses, showing how alignment and uniformity can be modulated by a hyperparameter.

While these are important properties, they do not characterize how CL algorithms organize samples from different classes in the embedding space. As a result, several papers [42, 44, 52, 53, 54, 55, 56, 57, 58, 36, 59] studied CL's ability to recover meaningful clusters and latent variables in the data. However, these results often rely on strong assumptions about the data and augmentations, such as assuming augmentations of the same sample are conditionally independent given their cluster identity (see, e.g., [42, 43, 44, 36]). In order to avoid such assumptions, [60] studies function classes that induce a similar bias towards preserving clusters without any assumption on the connectedness of augmentation sets. However, their analysis is focused on minimizing a spectral contrastive loss which serves as a surrogate for the more practically used InfoNCE loss. [45] instead proves that InfoNCE learns cluster-preserving embeddings by capping the representation class's capacity so that any function that tries to carve a semantic cluster must also split an image from its own augmentation set, making cluster integrity the loss-minimizing choice.

Another way to understand SSL is by examining its connection to supervised learning. For example, [61] showed that, in linear models, certain SSL objectives resembling VicReg are equivalent to supervised quadratic losses. Building on this perspective, we establish a bound between the contrastive loss and a supervised variant of it. Unlike previous work, our bound is both architecture-independent and label-agnostic.

Other theoretical studies have analyzed SSL from different angles, such as feature learning dynamics in linear models and two-layer networks [62, 63, 64, 65], the role of augmentations [12, 66], the projection head [67, 68, 69, 70], CL's sample complexity [71] and how to relax the dependence on large mini-batches [72]. Other work explores connections between self-supervised contrastive and non-contrastive learning [73, 74, 75, 76, 77].

**Neural collapse.** Neural collapse (NC) [29, 30] is a phenomenon that occurs in the final stages in training of overparameterized networks for classification. In this regime, we observe: (i) *within-class collapse*, where the embeddings of samples within each class converge to a single vector; (ii) *class separation*, where these class means spread out and often form a Simplex Equiangular Tight Frame (ETF); and a form of (iii) *alignment* between the feature space and the classifier's weights.

To understand the emergence of neural collapse, several papers studied the emergence of these properties in different learning settings. In [78, 79] they introduced the "unconstrained features model" or "layer peeled model"—where each sample's embedding is treated as a free parameter in $\mathbb{R}^d$. Under such relaxations, research has demonstrated that neural collapse is a characteristic of global minimizers for many commonly used supervised loss functions [80], including cross-entropy [31, 32, 33], mean squared error (MSE) [30, 34, 35], and supervised contrastive loss [33, 36, 37, 38]. In Thm. 2 we use a similar approach in order to characterize the global minima of the CL and NSCL losses, connecting them to downstream performance using tools from [46, 81, 82].

## 3 Theoretical Analysis

In this section, we explore *why* CL—despite being agnostic to class labels—tends to organize data by class and support few-shot transfer. We first relate the self-supervised loss to a label-aware loss function, then describe the geometric conditions under which a representation supports few-shot transfer, and finally show how this loss function induces these properties.

**Setup.** We consider training an embedding function $f \in \mathcal{F} \subset \{f \mid f : \mathcal{X} \to \mathbb{R}^d\}$ via self-supervised contrastive learning, using a dataset $S = \{(x_i, y_i)\}_{i=1}^N \subset \mathcal{X} \times [C]$, where the algorithm only sees inputs $x_i \in \mathcal{X}$ (e.g., images), but not the corresponding class labels $y_i \in [C]$. We assume that each class contains at most $n_{\max}$ samples. We wish to learn a "meaningful" (see Sec. 3.2) function $f : \mathcal{X} \to \mathbb{R}^d$ that maps samples to $d$-dimensional embedding vectors.

Concretely, for each sample $x_i$, we also construct $K$ *augmented* versions $x_i^l = \alpha_l(x_i)$ (via data augmentations $\alpha_1, \ldots, \alpha_K$, e.g. identity, random cropping, jitter, etc.). Then, we define $z_i^l = f(\alpha_l(x_i))$ as the embeddings of the augmented samples.

We focus on the global decoupled contrastive loss (DCL) [25], which is given by

$$\mathcal{L}^{\mathrm{DCL}}(f) \;=\; -\frac{1}{K^2 N} \sum_{l_1, l_2=1}^{K} \sum_{i=1}^{N} \log \left( \frac{\exp(\mathrm{sim}(z_i^{l_1}, z_i^{l_2}))}{\sum_{l_3=1}^{K} \sum_{j \in [N] \setminus \{i\}} \exp(\mathrm{sim}(z_i^{l_1}, z_j^{l_3}))} \right), \tag{1}$$

where $\mathrm{sim}(\cdot, \cdot) : \mathbb{R}^d \times \mathbb{R}^d \to \mathbb{R}$ is the cosine similarity function, which is defined as follows: $\mathrm{sim}(x_1, x_2) = \frac{\langle x_1, x_2 \rangle}{\|x_1\|_2 \cdot \|x_2\|_2}$. We focus on the DCL loss since its a slight improvement to the original global CL loss [2, 5, 27, 72, 5], where $\sum_{j=1}^N$ is replaced with $\sum_{j \neq i}^N$ in the denominator. While here we focus on the global [72, 49], for completeness, in Appendix B.1 we extend Thm. 1 to mini-batch CL losses as done in practice.

Despite the lack of label supervision, contrastive models often learn class-aware features. This is surprising: **Why should a label-agnostic loss promote semantic structure?**

### 3.1 Self-Supervised vs. Supervised Contrastive Learning

To answer this, we compare DCL with a supervised contrastive loss that removes same-class contrasts. Namely, we consider the *negatives-only supervised contrastive loss* (NSCL), $\mathcal{L}^{\mathrm{NSCL}}(f)$, which is defined the same as DCL in (1) but with $\sum_{j \in [N] \setminus \{i\}}$ in the denominator replaced by $\sum_{j : y_j \neq y_i}$.

When comparing the two losses, the unsupervised denominator includes at most extra $K(n_{\max} - 1)$ terms corresponding to data from the same class. Therefore, if $n_{\max}$ is small relative to the number of classes $C$ (i.e., when $C$ is large), the extra $K(n_{\max} - 1)$ terms are expected to be negligible compared to the total number of negatives $\geq K(N - n_{\max})$, and the two objectives, $\mathcal{L}^{\mathrm{DCL}}(f)$ and $\mathcal{L}^{\mathrm{NSCL}}(f)$, become nearly equivalent. For instance, assume for simplicity that $\exp(\mathrm{sim}(z_i, z_j)) \approx \gamma$ for all $i \neq j$ (i.e., the similarity values are approximately constant). Then the unsupervised denominator is approximately $K(N-1)\gamma$ while the supervised denominator (excluding same-class data) becomes approximately $K(N - n_{\max})\gamma$. Overall, we obtain that $|\mathcal{L}^{\mathrm{DCL}}(f) - \mathcal{L}^{\mathrm{NSCL}}(f)| \approx \log\left(1 + \frac{n_{\max}}{N - n_{\max}}\right) \leq \frac{n_{\max}}{N - n_{\max}}$. Hence, the gap between the two losses shrinks as $C$ grows.

In essence, this example shows that if $\exp(\mathrm{sim}(z_i^{l_1}, z_j^{l_2}))$ is nearly constant, removing same-class negatives has little effect on the loss. The following theorem shows that one can achieve a slightly worse bound on $|\mathcal{L}^{\mathrm{DCL}}(f) - \mathcal{L}^{\mathrm{NSCL}}(f)|$ without assuming that $\exp(\mathrm{sim}(z_i^{l_1}, z_j^{l_2}))$ is constant.

**Theorem 1.** *Let $S = \{(x_i, y_i)\}_{i=1}^N \subset \mathcal{X} \times [C]$ be a labeled dataset with $C$ classes, each containing at most $n_{\max}$ distinct samples. Let $f : \mathcal{X} \to \mathbb{R}^d$ be any function. Then, we have*

$$\mathcal{L}^{\mathrm{NSCL}}(f) \;\leq\; \mathcal{L}^{\mathrm{DCL}}(f) \;\leq\; \mathcal{L}^{\mathrm{NSCL}}(f) + \log\left(1 + \frac{n_{\max}\mathrm{e}^2}{N - n_{\max}}\right) \;\leq\; \mathcal{L}^{\mathrm{NSCL}}(f) + \frac{n_{\max}\mathrm{e}^2}{N - n_{\max}},$$

*where $\mathrm{e}$ denotes Euler's constant. For a balanced classification problem, $\frac{n_{\max}}{N - n_{\max}} = \frac{1}{C-1}$.*

The above theorem gives a justification for why contrastive SSL yields class-aware features without ever seeing the labels. Interestingly, since the right-hand side depends only on the number of samples $N$ and the maximal number of samples per class $n_{\max}$—not on the specific assignment $Y$—the inequality is label-agnostic. Namely, for any labeling $Y$ with $n_{\max}$ samples per class, we have,

$$0 \;\leq\; \mathcal{L}^{\mathrm{DCL}}(f) - \mathcal{L}^{\mathrm{NSCL}}_{(Y)}(f) \;\leq\; \log(1 + \tfrac{n_{\max}\mathrm{e}^2}{N - n_{\max}}), \tag{2}$$

where $\mathcal{L}^{\mathrm{NSCL}}_{(Y)}(f)$ is the NSCL loss under that labeling. In particular, $C \to \infty$ the term $\log(1 + \frac{n_{\max}\mathrm{e}^2}{N - n_{\max}})$ vanishes, the probability of drawing a same-class contrasting drops to zero, and DCL effectively collapses onto the NSCL for all labelings simultaneously.

## 3.2 Characterizing Good Representations for Few-Shot Learning

The primary goal of SSL is to learn representations that are easily adaptable to downstream tasks. We now describe the conditions under which an SSL-trained representation $f$ can be adapted to a specific downstream task. For this purpose, we consider the *recoverability* of labels from a learned representation $f : \mathcal{X} \to \mathbb{R}^d$. We call $f$ *"good"* if it supports accurate downstream classification with only a few labeled samples per class.

We formalize this via the expected $m$-shot classification error. Suppose we have a set of class-conditional distributions $D_1, \ldots, D_{C'}$ and define $D = \frac{1}{C'} \sum_{c=1}^{C'} D_c$ as their uniform mixture. For example, $D_1, \ldots, D_{C'}$ can represent a subset of $C' \le C$ classes from the pretraining task, where all $D_i$ are either training datasets or data distributions corresponding to their respective classes. Given an $m$-shot training set $\widehat{S}_i \sim D_i^m$ for each class $i$, we define: $\mathrm{err}_{m,D}(f) := \mathbb{E}_{\widehat{S}_1, \ldots \widehat{S}_{C'}}[\mathrm{err}_D(h_{f,\widehat{S}})]$, where $h_{f,\widehat{S}}$ is a classifier trained using only the small labeled support $\widehat{S} = \cup_{c=1}^{C'} \hat{S}_c$ and $\mathrm{err}_D(h)$ is the probability that $h$ misclassifies a ran-

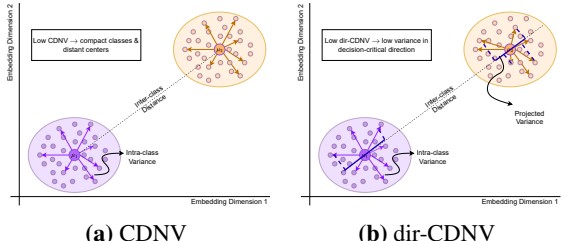

**(a)** CDNV      **(b)** dir-CDNV

Figure 2: **Illustration of CDNV and directional CDNV.** CDNV compares how tightly samples within a class cluster to how far apart class centers are; lower values indicate tighter clusters and larger gaps between classes. The latter measures variability only along the line connecting two class centers, highlighting the component most relevant for distinguishing those classes.

dom test point drawn from the task distribution $D$. Specifically, we denote linear probing (LP) and nearest-class-center classification (NCCC) errors by $\mathrm{err}^{\mathrm{LP}}_{m,D}(f)$ and $\mathrm{err}^{\mathrm{NCC}}_{m,D}(f)$ which refer to $\mathrm{err}_{m,D}(f)$ with $h_{f,\widehat{S}}$ being a linear classifier that minimizes $\mathrm{err}_{\widehat{S}}(h)$ and the nearest-class-center classifier $\arg \min_{c \in [C']} \|f(x) - \mu_f(\widehat{S}_c)\|_2$.

Prior work [46, 81, 82] shows that this error can be bounded in terms of how *clustered* the representations are. Specifically, for each class $i$, let $\mu_i = \mathbb{E}_{x \sim D_i}[f(x)]$ and $\sigma_i^2 = \mathbb{E}_{x \sim D_i}\big[\|f(x) - \mu_i\|_2^2\big]$ denote the mean and variance of the class embeddings, and let $V_f(D_i, D_j) = \sigma_i^2 / \|\mu_i - \mu_j\|^2$ be the class-distance-normalized variance [46, 81, 82] between classes $i$ and $j$.

These works (see, e.g., Prop. 7 in [82]) showed that

$$\mathrm{err}^{\mathrm{LP}}_{m,D}(f) \le \mathrm{err}^{\mathrm{NCC}}_{m,D}(f) \lesssim (C'-1)(1 + \tfrac{1}{m})\mathrm{Avg}_{i \ne j}[V_f(D_i, D_j)]. \tag{3}$$

In supervised learning, representations tend to become tightly clustered, causing CDNV to decrease, and thus reducing few-shot error [29, 46, 83, 84].

However, in the self-supervised case, labels are not available during training. As a result, there is no mechanism to explicitly encourage intra-class similarity, and thus we cannot generally expect that $V_f = \mathrm{Avg}_{i \ne j}[V_f(D_i, D_j)]$ will be small.

To better capture settings where label information is implicit, we introduce a refinement of (3) that prioritizes *directional variability* (see Fig. 2 for visual illustration). For each pair $i, j$, let: $\sigma_{ij}^2 = \mathrm{Var}_{x \sim D_i}\big[\langle f(x) - \mu_i, u_{ij}\rangle\big]$, where $u_{ij} = \frac{\mu_i - \mu_j}{\|\mu_i - \mu_j\|_2}$. Then the *directional* CDNV, $\tilde{V}_f = \mathrm{Avg}_{i \ne j}[\tilde{V}_f(D_i, D_j)]$, is defined as: $\tilde{V}_f(D_i, D_j) = \sigma_{ij}^2 / \|\mu_i - \mu_j\|_2^2$, that measures the variation along the line between the class centers. This leads to the following stronger bound:

**Proposition 1.** *Let $C' \ge 2$ and $m \ge 10$ be integers. Fix a feature map $f : \mathcal{X} \to \mathbb{R}^d$ and class-conditional distributions $D_1, \ldots, D_{C'}$ over $\mathcal{X}$. Let $V_f^s = \mathrm{Avg}_{i \ne j}[\sqrt{V_f(D_i, D_j)}]$. We have:*

$$\mathrm{err}^{\mathrm{LP}}_{m,D}(f) \le \mathrm{err}^{\mathrm{NCC}}_{m,D}(f) \le (C'-1)\left[8\tilde{V}_f + \tfrac{8}{\sqrt{m}}V_f^s + \tfrac{8}{\sqrt{m}}V_f + \tfrac{4}{m}V_f\right]. \tag{4}$$

Unlike (3), which depends on the full CDNV, Prop. 1 depends primarily on the directional CDNV, (and on $\frac{1}{m}V_f$), which is always smaller than the CDNV ($\tilde{V}_f \le V_f$) and in isotropic distributions $f \circ D_i$ it even scales as $\tilde{V}_f = \frac{1}{d}V_f$. Thus, the new bound can explain low $m$-shot error even when the regular CDNV is large. In essence, the bound predicts that SSL-trained models are effective

for downstream tasks when their directional CDNV is very low and their CDNV remains moderate. Fig. 7 shows that this prediction holds in practice.

Finally, we note that the coefficients (14 and 28.5) in the bound are sub-optimal. In (13) (Appendix C) we give a more general form of the coefficients, parameterized by a variable $a$. In Cor. 1 (Appendix C) we give an optimized version of the bound using a near-optimal choice of $a$. In Fig. 6(bottom), we show that the bound in Cor. 1 is fairly tight in practice.

## 3.3 Characterizing the representations learned with $\mathcal{L}^{\mathrm{NSCL}}$

Prop. 1 establishes that if the directional CDNV is small and the overall CDNV remains bounded, then the linear probing error of $f$ will likewise be small. Since Thm. 1 guarantees that minimizing the DCL loss also drives down the NSCL loss, we can use NSCL as a stand-in for DCL. Accordingly, in this section we analyze representations learned under the NSCL objective and evaluate how well they recover class labels through linear probing.

Because minimizing over all neural networks $f$ is intractable, we follow prior work [78, 79] and adopt the unconstrained features model. Specifically, we treat $f$ as an arbitrary function that given an augmented sample $x_i^l$ selects an embedding $z_i^l$ as a *free* learnable vector in $\mathbb{R}^d$. The following theorem characterizes the global minimizers of the NSCL loss, showing that they satisfy the NC1, NC2, and NC4 properties of NC [29, 30], and thus learn representations similar to those of minimizers of other classification losses, such as cross-entropy [31, 32, 33], MSE [30, 34, 35], and SCL [33].

**Theorem 2.** *Let $d \geq C - 1$ and let $S = \{(x_i, y_i)\}_{i=1}^N \subset \mathcal{X} \times [C]$ be a balanced labeled dataset with $C$ classes. Suppose $f$ is a global minimizer of the supervised contrastive loss $\mathcal{L}^{\mathrm{NSCL}}(f)$ (over all functions $f : \mathcal{X} \to \mathbb{R}^d$). Then, the representations satisfy the following properties:*

1. *Augmentation Collapse: For each $i \in [N]$ and for every pair $l_1, l_2 \in [K]$, we have $z_i^{l_1} = z_i^{l_2}$.*

2. *Within-Class Collapse: For any two samples $x_i$ and $x_j$ with the same label ($y_i = y_j$), their representations coincide: $z_i = z_j$. Namely, each class has a unique class embedding.*

3. *Simplex Equiangular Tight Frame: Let $\{\mu_1, \ldots, \mu_C\}$ denote the set of class-center embeddings. These vectors form a simplex ETF in $\mathbb{R}^d$; specifically, they satisfy $\sum_{c=1}^C \mu_c = 0$, $\|\mu_c\|_2 = \|\mu_{c'}\|_2$ and $\langle \mu_c, \mu_{c'} \rangle = -\frac{\|\mu_c\|_2^2}{C-1}$ for all $c \neq c' \in [C]$.*

Thm. 2 implies that any global minimizer of $\mathcal{L}^{\mathrm{NSCL}}(f)$ yields perfectly clustered representations. As such, the CDNV and the directional CDNV are zero and by Prop. 1, the 1-shot errors vanish: $\mathrm{err}_{m,S}^{\mathrm{LP}}(f) \leq \mathrm{err}_{m,S}^{\mathrm{NCC}}(f) = 0$. In CL on the other hand, we cannot achieve such collapse, since it would imply that the representations encode only label information–despite the loss being label-agnostic. Still, the DCL-NSCL duality invites a more nuanced question: **Does minimizing CL still promote weak forms of clustering?** While zero CDNV is tied to a specific labeling, the few-shot error $\mathrm{err}_{m,S}^{\mathrm{NCC}}(f)$ and the directional CDNV can be small under many possible labelings. In Sec. 4, we empirically explore the extent to which minimizing the DCL loss induces such a structure.

## 4 Experiments

### 4.1 Experimental Setup

**Datasets.** We experiment with the following datasets - CIFAR10 and CIFAR100 [85], mini-ImageNet [86], Tiny-ImageNet [87], SVHN [88] and ImageNet-1K [89].

**Methods, architectures and optimizers.** We trained our models with the SimCLR [5] algorithm. We use a ResNet-50 [90] encoder with a width-multiplier factor of 2 and ViT-Base [91]. Both are followed by a projection head with a standard two-layer MLP architecture composed of: `Linear(2048 → 2048)` → `ReLU` → `Linear(2048 → 128)`. For additional experiments and setup details about Vision Transformers [91] and MoCo v2 [6], see Appendix A.

Instead of training our models with the standard InfoNCE loss that is used in SimCLR, we use the DCL loss that avoids positive-negative coupling during training [25]. In order to minimize the loss, we adopt the LARS optimizer [92] which has been shown in [5] to be effective for training with large

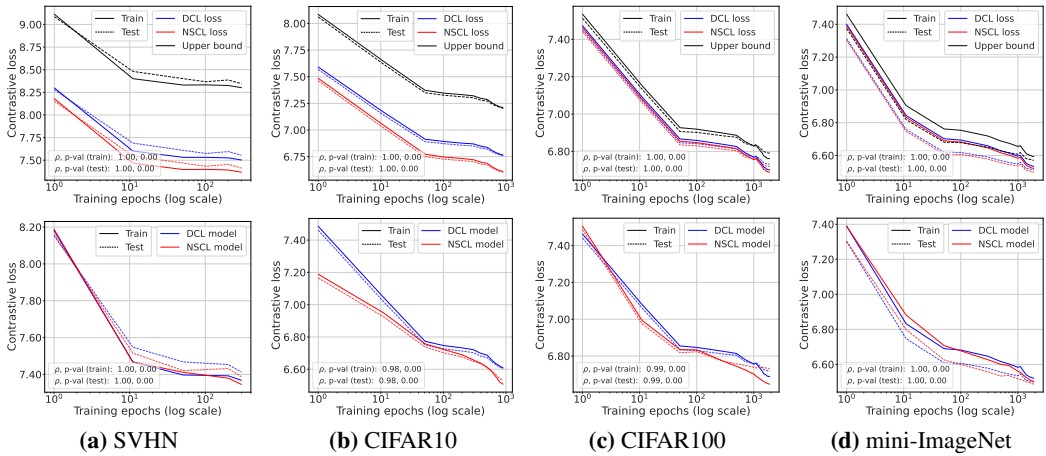

| (a) SVHN | (b) CIFAR10 | (c) CIFAR100 | (d) mini-ImageNet |

Figure 3: **(Top)** We train the model to minimize the DCL loss, tracking the DCL loss, the NSCL loss, and the bound NSCL+$\log(1 + \frac{n_{\max}\mathrm{e}^2}{N - n_{\max}})$ on both the training and test sets throughout training. *All three quantities are highly correlated.* **(Bottom)** We compare the NSCL loss of two models: one trained with the DCL loss and the other with the NSCL loss. *The resulting NSCL losses are comparable, regardless of the training objective.* In both the top and bottom plots, correlations are computed between the DCL and NSCL losses on the train and test data.

batch sizes. For LARS, we set the momentum to $0.9$ and the weight decay to $1\mathrm{e}^{-6}$. All experiments are carried out with a batch size of $B = 1024$. The base learning rate is scaled with batch size as $0.3 \cdot \lfloor B/256 \rfloor$, following standard practice [5]. We employ a warm-up phase [93] for the first 10 epochs, followed by a cosine learning rate schedule without restarts [94] for the remaining epochs. All models were trained on a single node with two 94 GB NVIDIA H100 GPUs.

**Evaluation metrics.** In several experiments we monitor $\mathcal{L}^{\mathrm{DCL}}$ (see (1)) and $\mathcal{L}^{\mathrm{NSCL}}$ (see 3.1). To calculate each one of these loss functions, we replace the sum over samples and their $K$ augmentations in the denominator with a sum over a random batch of $B = 1024$ random samples and one random augmentation for each sample as in SimCLR (see Appendix B.1 for details).

To evaluate the quality of learned representations, we use two methods: the Nearest Class-Center Classification (NCCC) accuracy [83], and linear probing accuracy [95, 96, 97]. Suppose we have a feature map $f$, a classification task with $C$ classes, and a dataset $S = \cup_{i=1}^{C} S_i$ (either training or test data). We estimate $\mathrm{err}_{m,S}^{\mathrm{LP}}(f)$ and $\mathrm{err}_{m,S}^{\mathrm{NCC}}(f)$ with $h_{f,\widehat{S}}$ being an NCC classifier or a linear classifier. To train the linear classifier, we use cross-entropy minimization for 500 epochs, with batch size $\min(|\widehat{S}|, 256)$, learning rate $3\mathrm{e}^{-4}$, weight decay $5\mathrm{e}^{-4}$, and momentum $0.9$. The expectation over the selection of $\widehat{S} = \cup_{i=1}^{C} \widehat{S}_i$ is estimated by averaging the error over 5 selections of $\widehat{S}$ from $S$.

We also estimate the bound in Prop. 1. As mentioned above, the coefficients in (4) are sub-optimal, so we instead use the refined bound from Cor. 1 (see Appendix C). To assess its ability to predict downstream performance, we compute the bound on both the training and test data by treating the per-class train/test subsets $S_i$ as the distributions $D_i$. The CDNV and the directional CDNV are calculated exactly as described in Sec. 3.2.

## 4.2 Experimental Results

**Validating Thm. 1 during training.** We train models using SimCLR to minimize the DCL loss. We evaluate both the DCL and NSCL losses on both training and test sets. Additionally, we evaluate the proposed upper bound, given by $\mathcal{L}^{\mathrm{NSCL}}(f) + \log(1 + \frac{n_{\max}\mathrm{e}^2}{N - n_{\max}})$, where $N$ is the total number of samples and $n_{\max}$ is the maximal number of samples per class (see Thm. 1).

In Fig. 3(top) we observe that the DCL loss consistently upper bounds the NSCL loss, and that the two losses become closer for tasks with a larger number of classes (e.g., CIFAR100 and mini-ImageNet compared to CIFAR10 and SVHN). In addition, when $C$ is large (e.g., $C = 100$), the bound becomes very tight, closely matching the losses. Al-

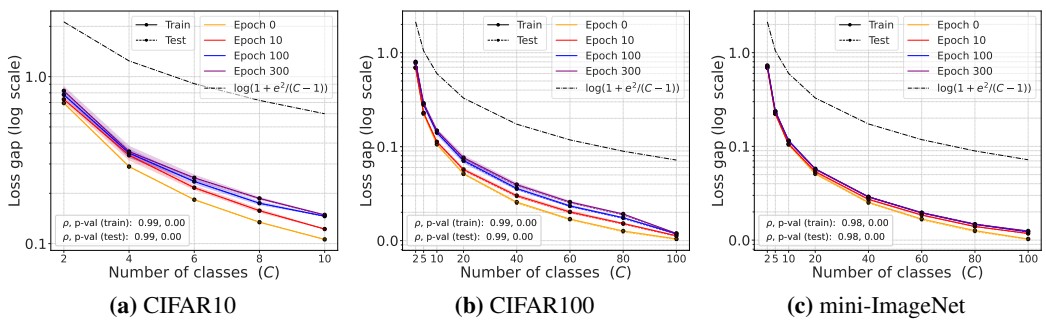

**(a)** CIFAR10      **(b)** CIFAR100      **(c)** mini-ImageNet

Figure 4: *The gap between the DCL and NSCL losses shrinks as the number of classes $C$ grows.* Models were trained to minimize the DCL loss, and at several training epochs we plot the empirical difference $\mathcal{L}^{\text{DCL}} - \mathcal{L}^{\text{NSCL}}$ alongside the bound $\log(1 + \frac{e^2}{C-1})$ as a function of $C$. We also report correlation between the loss gap at epoch 300 and the bound.

though we notice a slight divergence between the self-supervised and supervised loss metrics as training progresses, the overall trend continues to align well with the inequality stated in Thm. 1. For similar experiments with SimCLR-ViT and MoCo-v2, see Fig. 9 and Fig. 10.

While this result shows that for large $C$, the NSCL loss decreases alongside the DCL loss, it does not imply that minimizing the DCL loss leads to the same solu-

| Dataset | CIFAR10 | | CIFAR100 | | mini-ImageNet | |
| --- | --- | --- | --- | --- | --- | --- |
| | NCCC | LP | NCCC | LP | NCCC | LP |
| DCL | $85.3 \pm 2e^{-1}$ | $86.3 \pm 8e^{-3}$ | $57.3 \pm 1e^{-1}$ | $61.7 \pm 5e^{-2}$ | $69.0 \pm 2e^{-1}$ | $72.9 \pm 2e^{-2}$ |
| NSCL | $95.7 \pm 4e^{-3}$ | $95.6 \pm 4e^{-3}$ | $70.8 \pm 1e^{-1}$ | $73.7 \pm 2e^{-2}$ | $79.8 \pm 1e^{-1}$ | $81.3 \pm 2e^{-2}$ |

Table 1: NCCC and LP 100-shot test-time accuracy rates (and their standard deviations) of DCL and NSCL-trained models.

tion as directly minimizing the NSCL loss. To investigate this, we conducted an experiment in which we trained two models to minimize the DCL and NSCL losses (respectively) and compared their resulting NSCL values. As shown in Fig. 3(bottom), the gap between the NSCL losses of the two models is fairly small. This indicates that, in practice, optimizing the DCL loss leads to representations that are fairly clustered in comparison with those obtained by explicitly optimizing the NSCL loss.

**Validating Thm. 1 with varying $C$.** In the next experiment, we analyze how the gap $\mathcal{L}^{\text{DCL}}(f) - \mathcal{L}^{\text{NSCL}}(f)$ scales with the number of classes $C$. Specifically, we empirically validate that this gap is upper-bounded by $\log(1 + \frac{n_{\max}e^2}{N - n_{\max}}) = \log(1 + \frac{e^2}{C-1})$ (see Thm. 1), that the gap becomes tighter for large $C$, and that it is highly correlated with the actual gap between the losses.

For each value of $C$, we randomly sample $C$ classes from the full dataset, train a self-supervised model from scratch on data from these classes only, and compute $\mathcal{L}^{\text{DCL}}(f) - \mathcal{L}^{\text{NSCL}}(f)$ at various training epochs. To account for variability in class selection, we report the averaged value of this quantity over five independent random selections of $C$ classes along with error bars. Fig. 4 presents the empirical results, showing that the gap $\mathcal{L}^{\text{DCL}}(f) - \mathcal{L}^{\text{NSCL}}(f)$ slightly increases over the course of training. As shown in (5), the loss gap is given by $\log\left(1 + \frac{\sum_{j \neq i, y_j = y_i} \exp(\text{sim}(z_i, z_j))}{\sum_{y_j \neq y_i} \exp(\text{sim}(z_i, z_j))}\right)$. At initialization, the embeddings $z_i$ are randomly distributed in the representation space. During training, the embeddings cluster with respect to their classes thereby increasing $\text{sim}(z_i, z_j)$ for $y_j = y_i$ while reducing it for $y_j \neq y_i$. This leads to a gradual increase in the loss gap as observed in Fig. 4, but it remains consistently bounded by $\log(1 + \frac{e^2}{C-1})$ at all epochs. Moreover, the magnitude of this gap decreases with $C$ and is highly correlated with our bound.

**Comparing downstream performance.** To evaluate the quality of the learned representations, we measure the few-shot downstream performance of models trained with DCL and NSCL. Specifically, we report the NCCC error and linear probing error (see Sec. 4.1). Fig. 6 (top) shows train and test performance across all classes as a function of the number of shots per class ($m$). As can be seen, although the NCCC and linear probing errors of the DCL-trained models are, as expected, higher than those of the NSCL-trained model, they remain fairly low (see Table 1 for a numeric comparison). This

indicates that, despite not being explicitly optimized to align with class labels, the representations learned by DCL still exhibit strong clustering behavior.

Fig. 6 (bottom) compares the bound from Cor. 1 with the NCCC and linear-probe errors of DCL-trained models on 2-way downstream tasks (averaged over 10 tasks), for both train and test data. Unlike Prop. 7 in [82], and despite the constants in our bound, it is fairly tight as $m$ increases; for instance, at $m = 500$ for CIFAR10, it indicates that the few-shot errors are below 0.89. At $m = 10^6$, the test-time bound (red horizontal line) predicts an error of at most 0.26.

To complement our experiments, we take a publicly available SimCLR-ResNet-50 model pretrained on IM-1K[1], and perform 2-way $m$-shot NCCC evaluation similar to Fig. 6 (bottom). In Fig. 5, we verify our error bound with pretrained weights on mini-ImageNet and IM-1K.

Fig. 7 shows that DCL training only modestly lowers CDNV but reduces directional CDNV by about an order of magnitude during training. Because Prop. 1 links few-shot

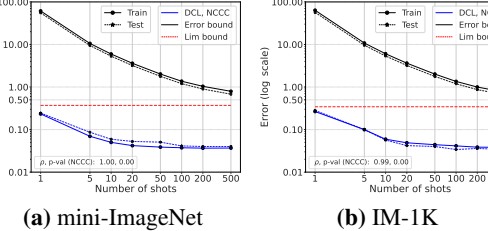

**(a)** mini-ImageNet    **(b)** IM-1K

Figure 5: **The bound in Cor. 1 is fairly tight for ImageNet pre-trained models.** We reproduced Fig. 6 with a ResNet-50 pre-trained on IM-1K.

error primarily to the directional CDNV, this sharp drop explains why DCL-trained models can recover labels (see Fig. 6) with limited supervision. In contrast, NSCL reduces both variances substantially, which accounts for its lower $m$-shot error.

**Augmentation collapse.** We verified augmentation collapse by training DCL ViT models for 2k epochs and measuring cosine similarity (0 indicates no collapse; 1 indicates perfect collapse). For each dataset, we computed the mean cosine similarity between augmentations of the same sample and between augmentations of different samples, using five augmentations per pair across the full dataset.

As shown in Tab. 2, same-sample augmentations achieve a mean cosine similarity of approximately 0.90, whereas different-sample pairs remain below 0.15. This consistent gap across CIFAR-10/100 and mini-ImageNet rules out trivial global collapse and indicates that DCL

| Dataset | CIFAR10 | | CIFAR100 | | mini-ImageNet | |
| --- | --- | --- | --- | --- | --- | --- |
| | same | diff | same | diff | same | diff |
| DCL | 0.92 | 0.13 | 0.91 | 0.11 | 0.89 | 0.07 |

Table 2: Cosine similarity between embeddings of two augmentations of the same sample ("positives") versus different samples ("negatives") for a ViT trained with DCL. Positives exhibit markedly higher similarity than negatives.

promotes strong augmentation invariance while preserving inter-sample separability.

**Representation Alignment.** Thm. 1 shows that the DCL and NSCL objectives have similar values, but this does not guarantee that the embeddings learned by minimizing the two objectives will have identical geometry. To better understand the structure of representations learned by models trained with DCL and NSCL, we run experiments that quantify representation similarity using Centered Kernel Alignment (CKA) [98] and Representation Similarity Analysis (RSA) [99]. When embeddings are highly aligned, RSA and CKA are close to 1; with no alignment, they approach 0.

We train DCL and NSCL models with the exact same initialization, same mini-batches, and same hyperparameter configurations as described in Sec. 4.1, for 300 epochs. As shown in Tab. 3, the models achieve very high RSA and CKA values that confirm alignment at the representation level.

| Encoder | CIFAR10 | | CIFAR100 | | mini-ImageNet | |
| --- | --- | --- | --- | --- | --- | --- |
| | RSA | CKA | RSA | CKA | RSA | CKA |
| ResNet-50 | 0.83 | 0.81 | 0.91 | 0.91 | 0.87 | 0.90 |
| ViT-Base | 0.86 | 0.82 | 0.91 | 0.91 | 0.89 | 0.89 |

Table 3: DCL and NSCL models trained for 300 epochs with matched initialization and minibatch order show very high representation similarity.

---

[1]https://github.com/AndrewAtanov/simclr-pytorch

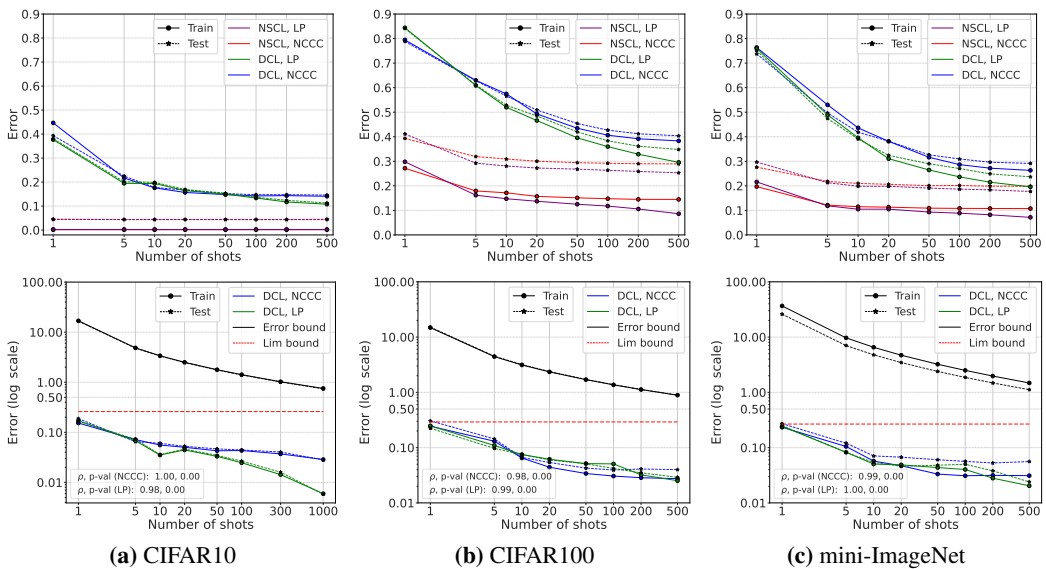

**(a)** CIFAR10  **(b)** CIFAR100  **(c)** mini-ImageNet

Figure 6: **(Top)** We compare the $m$-shot error (Linear Probing (LP) and Nearest Class-Centered Classifier (NCCC)) for $C$-way (all-class) classification. *Although the NCCC and LP errors of the DCL-trained models are, as expected, higher than those of the NSCL-trained model, they remain fairly low.* **(Bottom)** We compare the two-way $m$-shot NCCC and LP errors with the bound in Cor. 1. The errors are bounded by our bound, which decreases with $m$. The dashed red *'Lim bound'* line specifies the bound at $m = 10^6$, providing a tight test-time error estimate for large $m$. We also report correlation between the error bound and the errors $\mathrm{err}_{m,D}^{\mathrm{LP}}(f)$ and $\mathrm{err}_{m,D}^{\mathrm{NCCC}}(f)$ (on the test data).

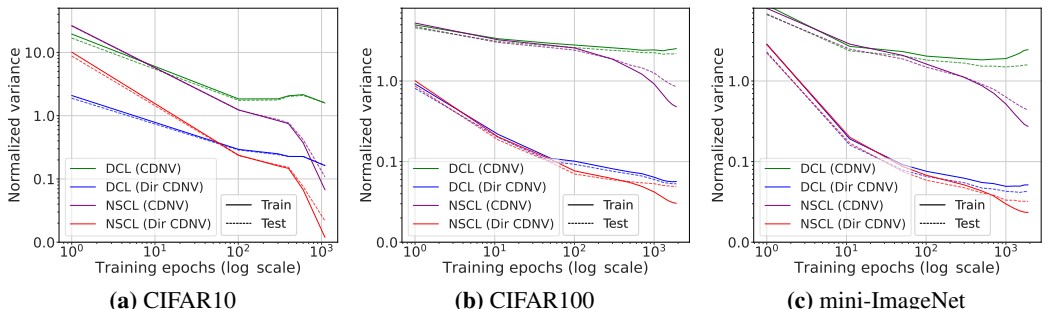

**(a)** CIFAR10  **(b)** CIFAR100  **(c)** mini-ImageNet

Figure 7: *DCL training yields a moderate reduction in CDNV and a significant reduction in directional CDNV, whereas NSCL training significantly decreases both.* We plot the CDNV and directional CDNV for both train and test data for DCL and NSCL-trained models over 2k epochs.

## 5 Discussion, Limitations, and Future Work

CL is at the forefront of modern pre-training techniques. Our work takes several steps toward a better understanding of CL by: establishing a duality between CL and NSCL, analyzing the minimizers of the NSCL loss, and linking the downstream error of CL-trained models to directional CDNV.

Nevertheless, our work has several important limitations that remain open for future investigation. While we show that the DCL and NSCL losses are close in value, this does not directly imply that their respective minimizers are close in parameter space or yield similar representations. It would be interesting to explore this connection both theoretically and empirically—for example, by monitoring the distance between two models trained using DCL and NSCL on the same batches. Another limitation is that the bound in Prop. 1 (and Cor. 1) involves large constants and scales linearly with $C'$, which may limit its practical utility. Tightening this bound by relaxing these dependencies would enhance its predictive power. Finally, extending these ideas to multimodal models such as CLIP [100], or to LLMs trained with analogous loss functions, presents an exciting direction for future research.

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

# A    Additional Experiments

**Datasets.**   We experiment with the following standard vision classification datasets - CIFAR10 and CIFAR100 [85], mini-ImageNet [86], Tiny-ImageNet [87], SVHN [88] and ImageNet-1K [89]. CIFAR10 and CIFAR100 both consist of 50000 training images and 10000 validation images with 10 classes and 100 classes, respectively, uniformly distributed across the dataset, i.e., CIFAR10 has 5000 samples per class and CIFAR100 has 500 samples per class. mini-ImageNet also has 5000 test images on top of 50000 train and 10000 validation images, with 100 of 1000 classes from ImageNet-1K [89] (at the original resolution). Tiny-ImageNet contains 100000 images downsampled to $64 \times 64$, with total 200 classes from IM-1K. Each class has 500 training, 50 validation, and 50 test images. SVHN consists of digit classification data with 10 classes from real-world images, organized into "train" (73,257 samples), "test" (26,032 samples), and "extra" (531,131 samples) splits. For scalability verification, we combine the "train" and "extra" splits during training for validating Thm. 1 (shown in Fig. 3). ImageNet-1K spans 1000 object classes and contains 1,281,167 training images, 50,000 validation images and 100,000 test images. We use "train" split for training and "validation" split for reporting test results.

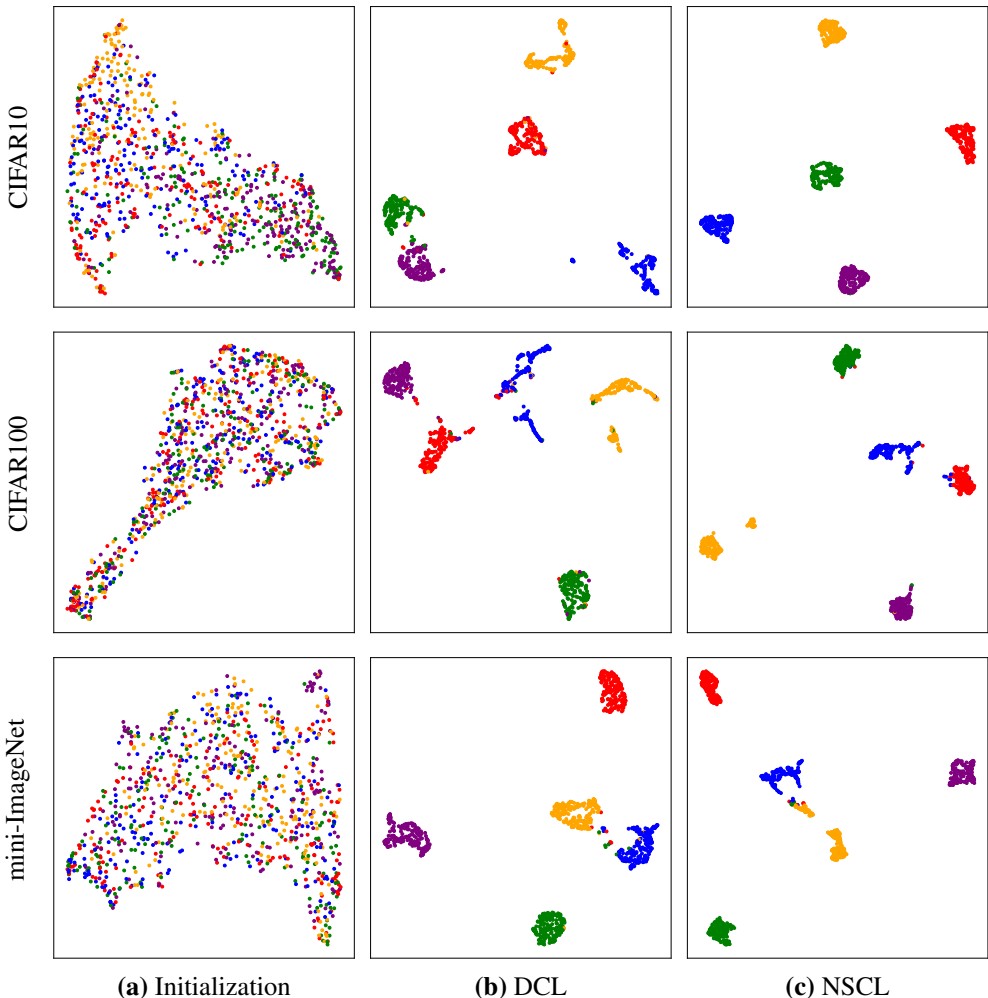

(a) Initialization        (b) DCL        (c) NSCL

Figure 8: UMAP visualizations of representations from models trained with different objectives (Random init, DCL, NSCL) on CIFAR10, CIFAR100, and mini-ImageNet. Each point corresponds to an image embedding colored by class. Better clustering and separation are evident as we move from Random to NSCL.

**Data augmentations.**   We use the same augmentations as in SimCLR [5]. For experiments on mini-ImageNet, and IM-1K, we use the following pipeline: random resized cropping to $224 \times 224$,

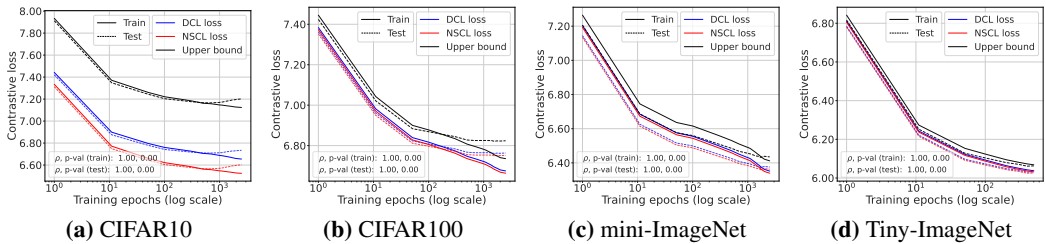

**(a)** CIFAR10      **(b)** CIFAR100      **(c)** mini-ImageNet      **(d)** Tiny-ImageNet

Figure 9: We reproduced **Fig. 3 (top)** for SimCLR with ViT-Base architecture. The loss-curves remain tightly bound as per Thm 1, independent of the encoder network architecture.

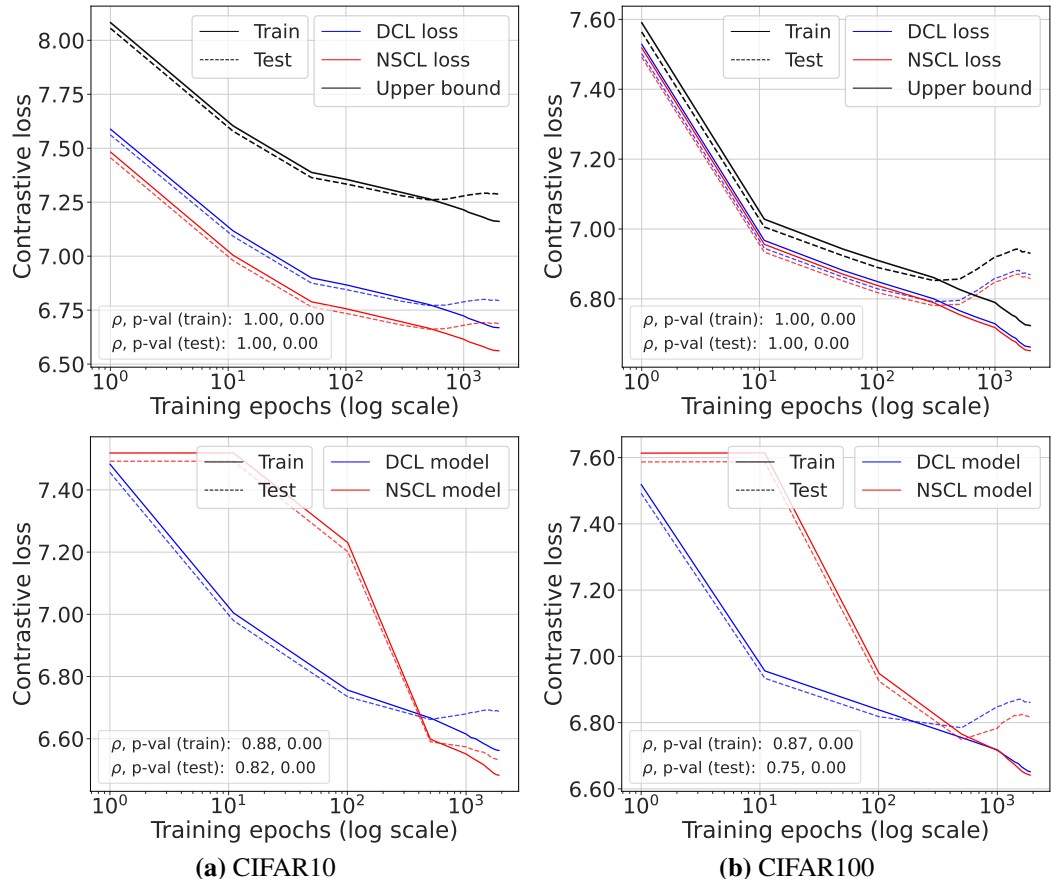

**(a)** CIFAR10          **(b)** CIFAR100

Figure 10: **The DCL and NSCL losses, along with our bound, for MoCo.** Same as Fig. 3 with MoCo training in place of SimCLR.

random horizontal flipping, color jittering (brightness, contrast, saturation: $0.8$; hue: $0.2$), random grayscale conversion ($p = 0.2$), and Gaussian blur (applied with probability $0.1$ using a $3 \times 3$ kernel and $\sigma = 1.5$). For CIFAR datasets and SVHN, we adopt a similar pipeline with appropriately scaled parameters. The crop size is adjusted to $32 \times 32$, and the color jitter parameters are scaled to saturation $0.4$, and hue $0.1$. For Tiny-ImageNet, we use the same saturation and hue values as CIFAR datasets with the crop size is adjusted to $64 \times 64$.

**UMAP visualizations of clustering behaviors.** In Fig. 1, we visualize the clustering behavior of DCL and NSCL-trained representations learned on mini-ImageNet using 2D UMAP [102] projections. We randomly selected 5 classes from mini-ImageNet and sampled 200 images per class, projecting their corresponding embeddings into 2D space using UMAP. As shown in Fig. 1, training with the NSCL loss results in tight and clearly separable clusters, as predicted in Thm. 2. Although DCL is label-agnostic, as shown in Thm. 1 and Fig. 3, DCL training implicitly minimizes the NSCL loss,

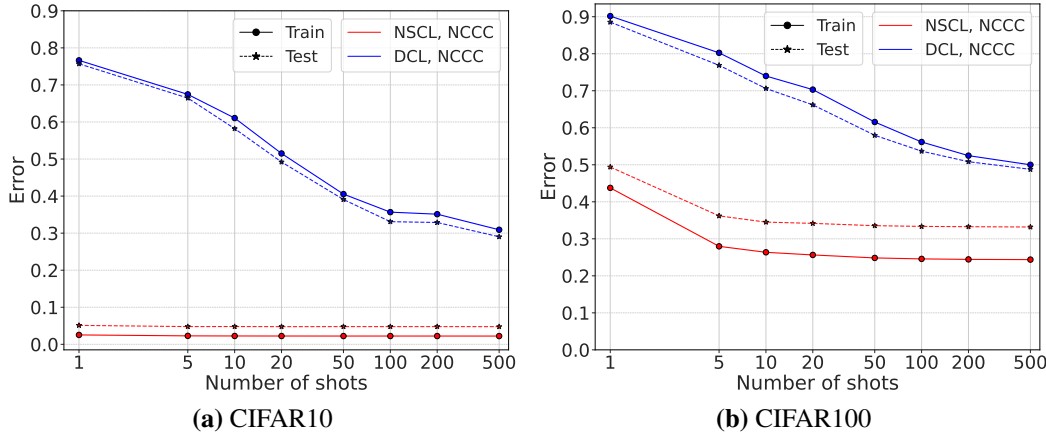

**(a)** CIFAR10          **(b)** CIFAR100

Figure 11: $m$-shot error (Nearest Class-Centered Classifier (NCCC)) for $C$-way (all-class) classification.

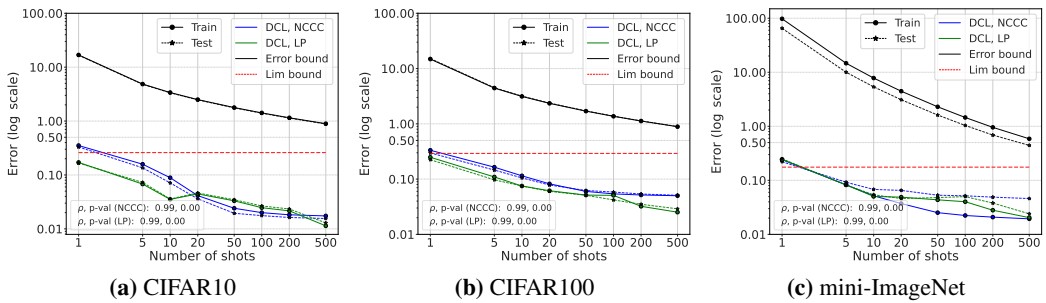

**(a)** CIFAR10      **(b)** CIFAR100      **(c)** mini-ImageNet

Figure 12: Two-way $m$-shot NCCC and LP errors with **ViT-Base** to verify the bound in Cor. 1, as shown in **Fig. 6 (bottom)**.

which explains the visible clustering behavior seen in Fig. 1. In Fig. 8, we demonstrate that the behavior generalizes across different datasets, $i.e.$, CIFAR10, CIFAR100, and mini-ImageNet.

## A.1 Extension to SimCLR (w/ ViT) and MoCo

To verify the generality of our empirical findings in the main text, we repeat some primary experiments with Vision Transformer (ViT) [91] in SimCLR, and also using the Momentum Contrast method, specifically MoCo v2 [6]. This section summarizes empirical results of SimCLR-ViT and MoCo, mirroring our analyses for SimCLR-ResNet-50 which we presented earlier.

**SimCLR-ViT setup.** We adopt the ViT-Base (ViT-B/16) architecture, which consists of 12 transformer layers, each with 12 attention heads and a hidden dimension of 768. For input images of size $224 \times 224$, we use a patch size of $16 \times 16$, resulting in 196 tokens (plus a [CLS] token). The MLP hidden dimension is 3072, and layer normalization is applied before each attention and MLP block. For CIFAR10 and CIFAR100, we follow the scaling procedure described in [103], adapting patch size and positional embeddings accordingly.

**MoCo setup.** We use the same architecture as with SimCLR. To train our model, we use SGD as an optimizer. We set momentum to 0.9 and the weight decay to $1e^{-4}$. All experiments are carried out with a batch size of $B = 256$. The base learning rate is set to 0.03 [27]. Similar to our SimCLR training strategy, we employ a warm-up phase [93] for the first 10 epochs, followed by a cosine learning rate schedule [94].

### A.1.1 Results

**Validating Thm. 1 during training.** We train SimCLR-ViT and MoCo using the DCL loss for 2k epochs, and evaluate both the DCL and NSCL losses on training and test datasets, along with the

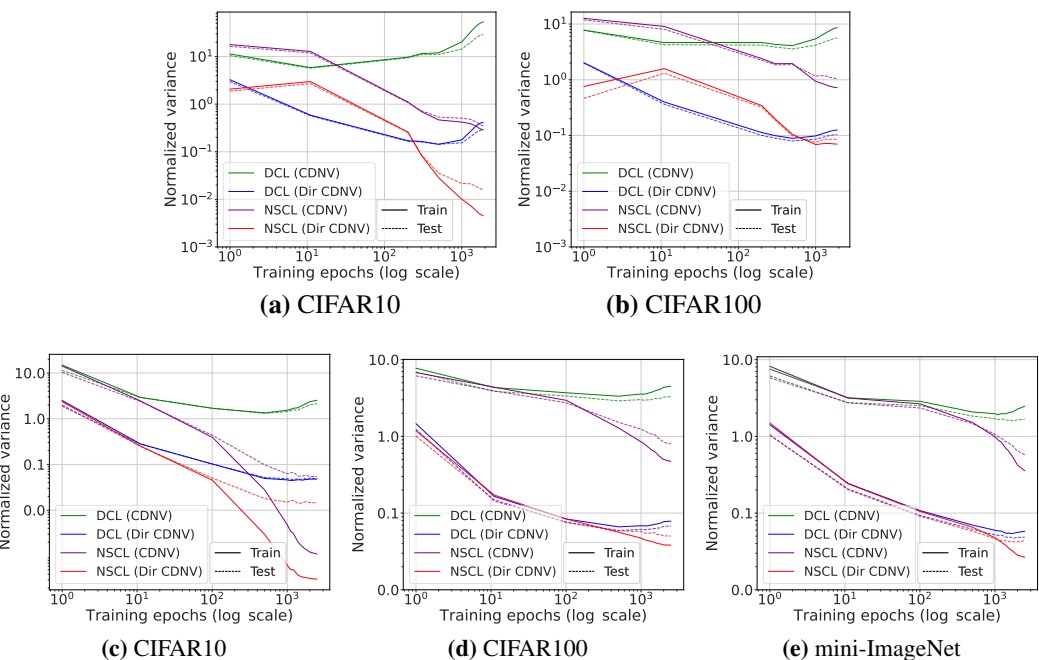

**(a)** CIFAR10  **(b)** CIFAR100

**(c)** CIFAR10  **(d)** CIFAR100  **(e)** mini-ImageNet

Figure 13: *CDNV and directional CDNV analysis.* **Top Row**: Results for a MoCo-trained ResNet-50 on CIFAR10 and CIFAR100. **Bottom Row**: Results for a SimCLR-trained ViT-Base on CIFAR10, CIFAR100, and mini-ImageNet.

theoretical upper bound $\mathcal{L}^{\text{NSCL}}(f) + \log(1 + \frac{n_{max}\mathrm{e}^2}{N-n_{max}})$. We repeat the experiments illustrated in Fig. 3(top). In Fig. 9 and Fig. 10(top), we notice similar trends as we have shown for SimCLR. The DCL loss consistently upper bounds the NSCL loss, with a narrowing gap with increase in number of classes $C$. The empirical $\mathcal{L}^{\text{CL}}$ and $\mathcal{L}^{\text{NSCL}}$ values remain tightly bounded by our proposed bound throughout training. For MoCo, we further validate that minimizing the DCL loss implicitly leads to low NSCL loss by repeating the experiments in Fig. 3(bottom). As shown in Fig. 10(bottom), $\mathcal{L}^{\text{NSCL}}$ for both DCL and NSCL-trained models remain close. This confirms our earlier conclusion that optimizing DCL implicitly reduces the NSCL loss.

**Comparing downstream performance.** We evaluate the quality of the learned representations from MoCo models via few-shot error analysis as earlier shown in Fig. 6. Specifically, we report the NCCC error on both train and test datasets. In Fig. 11(top) for MoCo, we evaluate NCCC error on the complete dataset as a function of number of shots per class ($m$) and show a comparison between DCL and NSCL-based models. In Fig. 12 for SimCLR-ViT, we perform 2-way classification (averaged over 10 tasks) and compare the NCCC error with the bound from Cor. 1. We notice that the theoretical bound remains tight for increasing $m$.

Similarly to our previous analysis, we also evaluate CDNV and directional CDNV in Fig. 13 for both DCL and NSCL-based MoCo and SimCLR-ViT models. We observe that while CDNV initially decreases for both models, the NSCL-trained models achieve substantially lower CDNV values as training progresses, whereas DCL-trained models maintain higher CDNV values. For directional CDNV, both the models achieve substantial reductions (an order of magnitude). For CIFAR10, directional CDNV for NSCL drops an additional order of magnitude compared to DCL.

## B  Proof of Thm. 1

**Theorem 1.** *Let $S = \{(x_i, y_i)\}_{i=1}^{N} \subset \mathcal{X} \times [C]$ be a labeled dataset with $C$ classes, each containing at most $n_{\max}$ distinct samples. Let $f : \mathcal{X} \to \mathbb{R}^d$ be any function. Then, we have*

$$\mathcal{L}^{\text{NSCL}}(f) \; \leq \; \mathcal{L}^{\text{DCL}}(f) \; \leq \; \mathcal{L}^{\text{NSCL}}(f) + \log\left(1 + \frac{n_{\max}\mathrm{e}^2}{N-n_{\max}}\right) \; \leq \; \mathcal{L}^{\text{NSCL}}(f) + \frac{n_{\max}\mathrm{e}^2}{N-n_{\max}},$$

*where* $\mathrm{e}$ *denotes Euler's constant. For a balanced classification problem,* $\frac{n_{\max}}{N-n_{\max}} = \frac{1}{C-1}$.

*Proof.* First, we show that $\mathcal{L}^{\mathrm{NSCL}}(f) \leq \mathcal{L}^{\mathrm{DCL}}(f)$. For each anchor $(i, l_1)$ (with $i \in \{1, \ldots, N\}$ and $l_1 \in \{1, \ldots, K\}$), define

$$Z_{\mathrm{neg}}^{i,l_1} = \sum_{l_3=1}^{K} \sum_{\substack{j=1 \\ y_j \neq y_i}}^{N} \exp(\mathrm{sim}(z_i^{l_1}, z_j^{l_3})), \quad Z_{\mathrm{pos}}^{i,l_1} = \sum_{l_3=1}^{K} \sum_{\substack{j=1 \\ y_j = y_i}}^{N} \exp(\mathrm{sim}(z_i^{l_1}, z_j^{l_3}))$$

and $Z_{\mathrm{pos \setminus self}}^{i,l_1} = \sum_{l_3=1}^{K} \sum_{\substack{j=1 \\ y_j = y_i, j \neq i}}^{N} \exp(\mathrm{sim}(z_i^{l_1}, z_j^{l_3})).$

By the definitions of the decoupled contrastive loss $\mathcal{L}^{\mathrm{DCL}}(f)$ and the negatives-only supervised contrastive loss $\mathcal{L}^{\mathrm{NSCL}}(f)$, we obtain

$$
\begin{aligned}
&\mathcal{L}^{\mathrm{DCL}}(f) - \mathcal{L}^{\mathrm{NSCL}}(f) \\
&= \frac{1}{K^2 N} \sum_{i=1}^{N} \sum_{l_1=1}^{K} \sum_{l_2=1}^{K} \left[ -\log\left( \frac{\exp(\mathrm{sim}(z_i^{l_1}, z_i^{l_2}))}{Z_{\mathrm{neg}}^{i,l_1} + Z_{\mathrm{pos \setminus self}}^{i,l_1}} \right) + \log\left( \frac{\exp(\mathrm{sim}(z_i^{l_1}, z_i^{l_2}))}{Z_{\mathrm{neg}}^{i,l_1}} \right) \right] \\
&= \frac{1}{K^2 N} \sum_{i=1}^{N} \sum_{l_1=1}^{K} \sum_{l_2=1}^{K} \log\left( \frac{Z_{\mathrm{neg}}^{i,l_1} + Z_{\mathrm{pos \setminus self}}^{i,l_1}}{Z_{\mathrm{neg}}^{i,l_1}} \right) \\
&= \frac{1}{N} \sum_{i=1}^{N} \log\left( 1 + \frac{Z_{\mathrm{pos \setminus self}}^{i,l_1}}{Z_{\mathrm{neg}}^{i,l_1}} \right).
\end{aligned} \tag{5}
$$

Since $\log(1 + x) \geq 0$ for all $x \geq 0$, it follows that $\mathcal{L}^{\mathrm{NSCL}}(f) \leq \mathcal{L}^{\mathrm{DCL}}(f)$. A similar argument shows that $\mathcal{L}^{\mathrm{DCL}}(f) \leq \mathcal{L}^{\mathrm{CL}}(f)$.

Next, by using the same reasoning as in (5) but replacing $Z_{\mathrm{pos \setminus self}}^{i,l_1}$ with $Z_{\mathrm{pos}}^{i,l_1}$, we have

$$\mathcal{L}^{\mathrm{CL}}(f) - \mathcal{L}^{\mathrm{NSCL}}(f) = \frac{1}{N} \sum_{i=1}^{N} \log\left( 1 + \frac{Z_{\mathrm{pos}}^{i,l_1}}{Z_{\mathrm{neg}}^{i,l_1}} \right).$$

Next, we bound the ratio $\frac{Z_{\mathrm{pos}}^{i,l_1}}{Z_{\mathrm{neg}}^{i,l_1}}$. For a fixed anchor $(i, l_1)$, note that there are at most $K n_{\max}$ positive terms (since each class contains at most $n_{\max}$ samples) and at least $K(N - n_{\max})$ negative terms. Moreover, because $\mathrm{sim}(z, z') \in [-1, 1]$, every term satisfies

$$\exp(-1) \leq \exp(\mathrm{sim}(z_i^{l_1}, z_j^{l_3})) \leq \exp(1).$$

Thus, we obtain

$$Z_{\mathrm{pos}}^{i,l_1} \leq K n_{\max} \exp(1) \quad \text{and} \quad Z_{\mathrm{neg}}^{i,l_1} \geq K(N - n_{\max}) \exp(-1).$$

Hence,

$$\frac{Z_{\mathrm{pos}}^{i,l_1}}{Z_{\mathrm{neg}}^{i,l_1}} \leq \frac{K n_{\max} \exp(1)}{K(N - n_{\max}) \exp(-1)} = \frac{n_{\max} \mathrm{e}^2}{N - n_{\max}}.$$

It follows that for each anchor,

$$\log\left( 1 + \frac{Z_{\mathrm{pos}}^{i,l_1}}{Z_{\mathrm{neg}}^{i,l_1}} \right) \leq \log\left( 1 + \frac{n_{\max} \mathrm{e}^2}{N - n_{\max}} \right).$$

Since this bound is uniform in $i$, $l_1$, and $l_2$, we conclude that

$$\mathcal{L}^{\mathrm{CL}}(f) - \mathcal{L}^{\mathrm{NSCL}}(f) \leq \log\left( 1 + \frac{n_{\max} \mathrm{e}^2}{N - n_{\max}} \right) \leq \frac{n_{\max} \mathrm{e}^2}{N - n_{\max}},$$

where the last inequality follows from $\log(1 + x) \leq x$ for all $x \geq 0$. $\qquad \square$

## B.1 Batch-Based Contrastive Losses

In practice, contrastive learning objectives are implemented using batches of samples rather than full-dataset sums [5]. To formalize this, let $B \in \mathbb{N}$ denote a chosen batch size. We assume access to a dataset $S = \{(x_i, y_i)\}_{i=1}^N \subset \mathcal{X} \times [C]$, where $x_i \in \mathcal{X}$ are inputs (e.g., images) and $y_i \in [C]$ are their class labels. For each sample $x_i$, let $x_i' \sim \alpha(x_i)$ be an independently generated augmentation from a distribution of augmentations $\alpha(x_i)$. Given a batch $\mathcal{B} = \{(x_{j_t}, x_{j_t}', y_{j_t})\}_{t=1}^B$ sampled with replacement from $S$, we define a per-sample contrastive loss for an anchor $(x_i, y_i)$ as:

$$\ell_{i,\mathcal{B}}(f) = -\log \left( \frac{\exp(\text{sim}(f(x_i), f(x_i')))}{\sum_{j \in \mathcal{B}} \left[ \exp(\text{sim}(f(x_i), f(x_j))) + \exp(\text{sim}(f(x_i), f(x_j'))) \right]} \right). \tag{6}$$

The self-supervised contrastive loss is then defined as the expectation over randomly chosen anchors, their augmentations and batches:

$$\mathcal{L}_B^{\text{CL}}(f) = \mathbb{E}_{x_i, x_i'} \mathbb{E}_{\mathcal{B}} \left[ \ell_{i,\mathcal{B}}(f) \right]. \tag{7}$$

In contrast, the negatives-only supervised contrastive loss restricts the negatives in each batch to only include examples from different classes. Specifically, for each anchor $(x_i, y_i)$, define $\mathcal{B}_i' = \{(x_{j_t}, x_{j_t}', y_{j_t})\}_{t=1}^B$ as a batch of size $B$, drawn (with replacement) from $S \setminus \{(x_j, y_j) : y_j = y_i\}$. Then, the loss is:

$$\mathcal{L}_B^{\text{NSCL}}(f) = \mathbb{E}_{x_i, x_i'} \mathbb{E}_{\mathcal{B}_i} \left[ \ell_{i,\mathcal{B}_i'}(f) \right].$$

With these notations we are ready to describe a bound on the gap between these two loss functions.

**Theorem 3.** *Let $S = \{(x_i, y_i)\}_{i=1}^N \subset \mathcal{X} \times [C]$ be a labeled dataset with $C$ classes, each containing $n$ distinct samples $(N = Cn)$. Let $B \in \mathbb{N}$ be the batch size, $\epsilon > 0$ be a positive number and $\bar{B} = \lceil B(1 - \frac{1}{C} - \epsilon) \rceil$. Let $f : \mathcal{X} \to \mathbb{R}^d$ be any function. Then, the difference between the self-supervised contrastive loss $\mathcal{L}_B^{\text{CL}}(f)$ and the decoupled supervised contrastive loss $\mathcal{L}_{\bar{B}}^{\text{NSCL}}(f)$ satisfies*

$$-2(\log(2B)+2)\exp(-2B\epsilon^2) \leq \mathcal{L}_B^{\text{CL}}(f) - \mathcal{L}_{\bar{B}}^{\text{NSCL}}(f) \leq \frac{e^2(1+\epsilon C)}{C(1-\epsilon)-1} + 2(\log(2B)+2)\exp(-2B\epsilon^2).$$

*Proof.* Fix an arbitrary anchor sample $(x_i, y_i) \in S$ and let $x_i'$ be an augmentation of $x_i$. Denote

$$z_i = f(x_i) \quad \text{and} \quad z_i' = f(x_i')$$

the corresponding embeddings. Next, let $\mathcal{B} = \{(x_{j_t}, x_{j_t}', y_{j_t})\}_{t=1}^B$ be a random batch of $B$ samples (with replacement) from $S$ along with their augmentations. For the anchor $(x_i, y_i)$, define the per-sample contrastive loss by

$$\ell_{i,\mathcal{B}}(f) = -\log \left( \frac{\exp\left(\text{sim}(z_i, z_i')\right)}{\sum_{j \in \mathcal{B}} \left[ \exp\left(\text{sim}(z_i, z_j)\right) + \exp\left(\text{sim}(z_i, z_j')\right) \right]} \right).$$

Thus, the overall self-supervised contrastive loss is

$$\mathcal{L}_B^{\text{CL}}(f) = \mathbb{E}_i \mathbb{E}_{\mathcal{B}} \left[ \ell_{i,\mathcal{B}}(f) \right].$$

For each anchor $i$, consider instead a batch

$$\mathcal{B}_i' = \{(x_{j_t}, x_{j_t}', y_{j_t})\}_{t=1}^{\bar{B}}$$

of $\bar{B} = \lceil B(1 - \frac{1}{C} - \epsilon) \rceil$ samples drawn (with replacement) from $S \setminus \{(x_j, y_j) : y_j = y_i\}$, i.e., only from the negatives relative to $i$. The decoupled supervised contrastive loss is defined as

$$\mathcal{L}_{\bar{B}}^{\text{NSCL}}(f) = \mathbb{E}_i \mathbb{E}_{\mathcal{B}_i'} \left[ \ell_{i,\mathcal{B}_i'}(f) \right].$$

For each anchor $i$, define the event

$$A_i = \left\{ \mathcal{B} : \#\{j \in \mathcal{B} : y_j \neq y_i\} \geq \bar{B} \right\}.$$

When $A_i$ holds, the distribution of a random subset $\mathcal{B}_i''$ of $\bar{B}$ negatives from $\mathcal{B}$ coincides with that of $\mathcal{B}_i'$. By the law of total expectation,

$$\begin{aligned}
\mathcal{L}_B^{\text{CL}}(f) - \mathcal{L}_{\bar{B}}^{\text{NSCL}}(f) &= \mathbb{E}_i \left[ \mathbb{P}[A_i] \, \mathbb{E}_{\mathcal{B}, \mathcal{B}_i'' | A_i} \left[ \ell_{i,\mathcal{B}}(f) - \ell_{i,\mathcal{B}_i''}(f) \right] \right] \\
&\quad + \mathbb{E}_i \left[ \mathbb{P}[\bar{A}_i] \left( \mathbb{E}_{\mathcal{B} | \bar{A}_i} \left[ \ell_{i,\mathcal{B}}(f) \right] - \mathbb{E}_{\mathcal{B}_i'} \left[ \ell_{i,\mathcal{B}_i'}(f) \right] \right) \right].
\end{aligned}$$

To bound the difference on the event $A_i$, define
$$Z_{i,\mathcal{B}} \;=\; \sum_{j \in \mathcal{B}} \left[ \exp\left(\mathrm{sim}(z_i, z_j)\right) + \exp\left(\mathrm{sim}(z_i, z_j')\right)\right].$$

Then, one may write
$$\ell_{i,\mathcal{B}}(f) - \ell_{i,\mathcal{B}_i''}(f) \;=\; \log\left(\frac{Z_{i,\mathcal{B}}}{Z_{i,\mathcal{B}_i''}}\right) \;=\; \log\left(1 + \frac{Z_{i,\mathcal{B}\setminus\mathcal{B}_i''}}{Z_{i,\mathcal{B}_i''}}\right).$$

Since $\log(1+u) \leq u$ for all $u \geq 0$, it follows that
$$\ell_{i,\mathcal{B}}(f) - \ell_{i,\mathcal{B}_i''}(f) \;\leq\; \frac{Z_{i,\mathcal{B}\setminus\mathcal{B}_i''}}{Z_{i,\mathcal{B}_i''}}.$$

Observe that $Z_{i,\mathcal{B}\setminus\mathcal{B}_i''}$ is a sum of $B - \bar{B}$ terms and, since $\mathrm{sim}(\cdot,\cdot) \in [-1,1]$, each term is bounded above by e. Similarly, $Z_{i,\mathcal{B}_i''}$ is a sum of $\bar{B}$ terms, each bounded below by $\mathrm{e}^{-1}$. Therefore,
$$\frac{Z_{i,\mathcal{B}\setminus\mathcal{B}_i''}}{Z_{i,\mathcal{B}_i''}} \;\leq\; \frac{\mathrm{e}(B - \bar{B})}{\mathrm{e}^{-1}\bar{B}} \;=\; \frac{\mathrm{e}^2(B - \bar{B})}{\bar{B}} \;\leq\; \frac{\mathrm{e}^2(B - B(1 - \frac{1}{C} - \epsilon))}{B\left(1 - \frac{1}{C} - \epsilon\right)} \;=\; \frac{\mathrm{e}^2(\frac{1}{C} + \epsilon)}{1 - \frac{1}{C} - \epsilon} \;=\; \frac{\mathrm{e}^2(1 + \epsilon C)}{C(1 - \epsilon) - 1}.$$

Thus, we deduce
$$\mathbb{E}_{\mathcal{B},\mathcal{B}_i''|A_i} \left[\ell_{i,\mathcal{B}}(f) - \ell_{i,\mathcal{B}_i''}(f)\right] \;\leq\; \frac{\mathrm{e}^2(1 + \epsilon C)}{C(1 - \epsilon) - 1}.$$

On the complement $\bar{A}_i$ the batch $\mathcal{B}$ contains fewer than $\bar{B}$ negatives. In this case one may bound the losses uniformly. In fact, using
$$\log\left(\sum_{j=1}^{B} \exp(\alpha_j)\right) \;\leq\; \max\{\alpha_1, \ldots, \alpha_B\} + \log(B)$$

and the fact that $\mathrm{sim}(\cdot,\cdot) \in [-1,1]$, we obtain
$$|\ell_{i,\mathcal{B}}(f)| \;\leq\; |\mathrm{sim}(z_i, z_i')| + \log\left(\sum_{j\in\mathcal{B}} \exp(\mathrm{sim}(z_i, z_j)) + \exp(\mathrm{sim}(z_i, z_j'))\right) \;\leq\; \log(2B) + 2,$$

and similarly,
$$\left|\ell_{i,\mathcal{B}_i'}(f)\right| \;\leq\; \log(2\bar{B}) + 2 \;\leq\; \log(2B) + 2.$$

Define the indicator variable $Y_j = \mathbf{1}[y_j \neq y_i]$ so that $\mathbb{E}[Y_j] = 1 - \frac{1}{C}$. By Hoeffding's inequality,
$$\mathbb{P}\left[\sum_{j=1}^{B} Y_j \leq \lceil B(1 - \tfrac{1}{C}) - B\epsilon\rceil\right] \;=\; \mathbb{P}\left[\sum_{j=1}^{B} Y_j \leq B(1 - \tfrac{1}{C}) - B\epsilon\right] \;\leq\; \exp(-2B\epsilon^2).$$

Hence, $\mathbb{P}[\bar{A}_i] \leq \exp\left(-B\epsilon^2\right)$ and the contribution of the event $\bar{A}_i$ is bounded by
$$\mathbb{E}_i \left[\mathbb{P}[\bar{A}_i] \cdot \left(\mathbb{E}_{\mathcal{B}|\bar{A}_i}\left[\ell_{i,\mathcal{B}}(f)\right] - \mathbb{E}_{\mathcal{B}_i''}\left[\ell_{i,\mathcal{B}_i''}(f)\right]\right)\right]$$
$$\leq\; \mathbb{E}_i \left[\mathbb{P}[\bar{A}_i] \cdot \left(\mathbb{E}_{\mathcal{B}|\bar{A}_i}\left[|\ell_{i,\mathcal{B}}(f)|\right] + \mathbb{E}_{\mathcal{B}_i''}\left[|\ell_{i,\mathcal{B}_i''}(f)|\right]\right)\right]$$
$$\leq\; 2\left(\log(2B) + 2\right)\exp\left(-B\epsilon^2\right).$$

Combining the bounds on $A_i$ and $\bar{A}_i$, we conclude that
$$\mathcal{L}_B^{\mathrm{CL}}(f) - \mathcal{L}_{\bar{B}}^{\mathrm{NSCL}}(f) \;\leq\; \frac{\mathrm{e}^2(1 + \epsilon C)}{C(1 - \epsilon) - 1} + 2\left(\log(2B) + 2\right)\exp\left(-2B\epsilon^2\right).$$

Finally, we note that under $A_i$, $\ell_{i,\mathcal{B}}(f) \geq \ell_{i,\mathcal{B}_i''}(f)$. In addition, similar to the above:
$$\mathbb{E}_i \left[\mathbb{P}[\bar{A}_i] \cdot \left(\mathbb{E}_{\mathcal{B}|\bar{A}_i}\left[\ell_{i,\mathcal{B}}(f)\right] - \mathbb{E}_{\mathcal{B}_i''}\left[\ell_{i,\mathcal{B}_i''}(f)\right]\right)\right] \;\geq\; -2\left(\log(2B) + 2\right)\exp\left(-B\epsilon^2\right).$$

yields the corresponding lower bound:
$$\mathcal{L}_B^{\mathrm{CL}}(f) - \mathcal{L}_{\bar{B}}^{\mathrm{NSCL}}(f) \;\geq\; -2\left(\log(2B) + 2\right)\exp\left(-2B\epsilon^2\right).$$

This completes the proof. $\qquad\square$

## C  Proof of Prop. 1

**Proposition 1.** *Let $C' \geq 2$ and $m \geq 10$ be integers. Fix a feature map $f : \mathcal{X} \to \mathbb{R}^d$ and class-conditional distributions $D_1, \ldots, D_{C'}$ over $\mathcal{X}$. Let $V_f^s = \mathrm{Avg}_{i \neq j}[\sqrt{V_f(D_i, D_j)}]$. We have:*

$$\mathrm{err}_{m,D}^{\mathrm{LP}}(f) \ \leq \ \mathrm{err}_{m,D}^{\mathrm{NCC}}(f) \ \leq \ (C'-1) \left[ 8\tilde{V}_f + \tfrac{8}{\sqrt{m}} V_f^s + \tfrac{8}{\sqrt{m}} V_f + \tfrac{4}{m} V_f \right]. \tag{4}$$

*Proof.* We adapt Prop. 7 of [82]. Assume $d_{ij} := \|\mu_j - \mu_i\|_2 > 0$ for all $i \neq j$ so that $u_{ij} := (\mu_j - \mu_i)/d_{ij}$ is well defined. Fix $i \neq j$. Draw independent samples $\widehat{S}_i = \{x_{i,1}, \ldots, x_{i,m}\} \sim D_i^m$ and $\widehat{S}_j = \{x_{j,1}, \ldots, x_{j,m}\} \sim D_j^m$, independent also of a fresh $x_i \sim D_i$. Let $\widehat{\mu}_c = \frac{1}{m} \sum_{s=1}^m f(x_{c,s})$ and $\mu_c = \mathbb{E}[f(x) \mid x \sim D_c]$ for $c \in \{i, j\}$, and write $z_i = f(x_i)$. Set $\delta_c := \widehat{\mu}_c - \mu_c$. For $c \in \{i, j\}$ define $z_c := f(x_c)$ with $x_c \sim D_c$, let $\Sigma_c := \mathrm{Cov}(z_c)$ and $v_c := \mathrm{tr}(\Sigma_c)$, and put

$$\tilde{V}_f(D_i, D_j) := \frac{u_{ij}^\top \Sigma_i u_{ij}}{d_{ij}^2}, \qquad V_f(D_i, D_j) := \frac{v_i + v_j}{d_{ij}^2}.$$

Let $D := \mu_j - \mu_i$, $d := \|D\|_2$, and $A := z_i - \frac{\mu_i + \mu_j}{2}$. A direct expansion gives

$$\|z_i - \widehat{\mu}_j\|^2 - \|z_i - \widehat{\mu}_i\|^2 = d^2 - 2dX \ - \ 2A^\top(\delta_j - \delta_i) \ + \ D^\top(\delta_j + \delta_i) \ + \ \left( \|\delta_j\|^2 - \|\delta_i\|^2 \right),$$

where $X := (z_i - \mu_i)^\top u_{ij}$. Hence NCC predicts $j$ iff

$$2X \ \geq \ d - \frac{2}{d}T + \frac{1}{d}S + \frac{1}{d}Q, \quad T := A^\top(\delta_j - \delta_i), \ S := D^\top(\delta_j + \delta_i), \ Q := \|\delta_j\|^2 - \|\delta_i\|^2. \tag{8}$$

Fix $a \geq 5$ and $m \geq 6$, and set $\tau(a, m) := \frac{1}{2} - \frac{2}{a} - \frac{2^{3/2}}{am} \in (0, 1/2)$, so $1 - 2\tau = \frac{4}{a} + \frac{2^{5/2}}{am}$. If

$$\frac{2|T|}{d^2} + \frac{|S|}{d^2} + \frac{|Q|}{d^2} \ \leq \ 1 - 2\tau, \tag{9}$$

then (8) implies $X \geq \tau d$, hence

$$\{\text{NCC predicts } j\} \ \subseteq \ \{X \geq \tau d\} \ \cup \ \left\{ \tfrac{2|T|}{d^2} + \tfrac{|S|}{d^2} + \tfrac{|Q|}{d^2} > 1 - 2\tau \right\}.$$

Since $\mathbb{E}[X] = 0$ and $\mathrm{Var}(X) = u_{ij}^\top \Sigma_i u_{ij}$,

$$\Pr[X \geq \tau d] \ \leq \ \frac{\mathrm{Var}(X)}{\tau^2 d^2} \ = \ \tau(a, m)^{-2} \, \tilde{V}_f(D_i, D_j). \tag{10}$$

By Markov and Cauchy–Schwarz,

$$\Pr\left[ \tfrac{2|T|}{d^2} + \tfrac{|S|}{d^2} + \tfrac{|Q|}{d^2} > 1 - 2\tau \right] \ \leq \ \frac{2\,\mathbb{E}|T| + \mathbb{E}|S| + \mathbb{E}|Q|}{(1 - 2\tau)d^2}. \tag{11}$$

Using independence of $(z_i, \delta_i, \delta_j)$ and $\mathbb{E}[\delta_c] = 0$,

$$\mathbb{E}[T^2] = \frac{1}{m} \left( \mathrm{tr}\left( \Sigma_i(\Sigma_i + \Sigma_j) \right) + \tfrac{d^2}{4} u_{ij}^\top (\Sigma_i + \Sigma_j) u_{ij} \right), \qquad \mathbb{E}[S^2] = \frac{d^2}{m} u_{ij}^\top (\Sigma_i + \Sigma_j) u_{ij}.$$

Since $\mathrm{tr}(AB) \leq \mathrm{tr}(A)\,\mathrm{tr}(B)$ for PSD $A$, $B$ and $u^\top \Sigma u \leq \mathrm{tr}(\Sigma)$,

$$\mathrm{tr}\left( \Sigma_i(\Sigma_i + \Sigma_j) \right) \ \leq \ v_i\,(v_i + v_j), \qquad u_{ij}^\top(\Sigma_i + \Sigma_j)u_{ij} \ \leq \ v_i + v_j.$$

Hence, by Cauchy–Schwarz and $\sqrt{ab} \leq (a + b)/2$,

$$\frac{2\,\mathbb{E}|T|}{d^2} \leq \frac{2}{\sqrt{m}\,d^2} \sqrt{v_i(v_i + v_j) + \frac{d^2}{4}(v_i + v_j)}$$

$$\leq \frac{1}{\sqrt{m}} \left( \sqrt{\frac{v_i + v_j}{d^2}} + 2\frac{v_i + v_j}{d^2} \right) = \frac{1}{\sqrt{m}} \left( \sqrt{V_{ij}} + 2V_{ij} \right),$$

and

$$\frac{\mathbb{E}|S|}{d^2} \leq \frac{1}{\sqrt{m}}\sqrt{\frac{v_i + v_j}{d^2}} = \frac{1}{\sqrt{m}}\sqrt{V_{ij}}, \qquad \mathbb{E}|Q| \leq \frac{v_i + v_j}{m} = \frac{V_{ij}\,d2}{m}.$$

Therefore

$$\frac{2\,\mathbb{E}|T| + \mathbb{E}|S| + \mathbb{E}|Q|}{d^2} \leq \frac{1}{\sqrt{m}}\big(2\sqrt{V_{ij}} + 2V_{ij}\big) + \frac{1}{m}V_{ij}. \tag{12}$$

Combining the union bound, (10) and (12),

$$\Pr(\text{NCC predicts } j) \leq \tau(a,m)^{-2}\,\tilde{V}_{ij} + \frac{1}{1 - 2\tau(a,m)}\left(\frac{2}{\sqrt{m}}\sqrt{V_{ij}} + \frac{2}{\sqrt{m}}V_{ij} + \frac{1}{m}V_{ij}\right).$$

Averaging over $i$ and summing over $j \neq i$ yields

$$\text{err}^{\text{NCC}}_{m,D}(f) \leq (C' - 1)\left[\tau(a,m)^{-2}\,\tilde{V}_f + \frac{1}{1 - 2\tau(a,m)}\left(\frac{2}{\sqrt{m}}V_f^s + \frac{2}{\sqrt{m}}V_f + \frac{1}{m}V_f\right)\right],$$

Since $(1 - 2\tau)^{-1} \leq a/4$, we obtain the simplified bound

$$\text{err}^{\text{NCC}}_{m,D}(f) \leq (C' - 1)\inf_{a \geq 5}\left[\left(\tfrac{1}{2} - \tfrac{2}{a} - \tfrac{2^{3/2}}{am}\right)^{-2}\tilde{V}_f + \frac{a}{4}\left(\frac{2}{\sqrt{m}}V_f^s + \frac{2}{\sqrt{m}}V_f + \frac{1}{m}V_f\right)\right]. \tag{13}$$

By picking $a = 16$ and using the assumption that $m \geq 10$ we get the desired bound. Finally, $\text{err}^{\text{LP}}_{m,D} \leq \text{err}^{\text{NCC}}_{m,D}$ since the Euclidean nearest-centroid rule is representable by a multiclass linear classifier. $\qquad\square$

**Corollary 1.** *Let $C' \geq 2$ and $m \geq 10$. Fix a feature map $f : \mathcal{X} \to \mathbb{R}^d$ and class-conditional distributions $D_1, \ldots, D_{C'}$ over $\mathcal{X}$. Write $\tilde{V}_f = \text{Avg}_{i \neq j}\big[\tilde{V}_f(D_i, D_j)\big]$, $V_f = \text{Avg}_{i \neq j}\big[V_f(D_i, D_j)\big]$, and $V_f^s = \text{Avg}_{i \neq j}\big[V_f^s(D_i, D_j)\big]$. Set*

$$A := 2 + \frac{2^{3/2}}{m}, \qquad B := \frac{1}{4}\left(\frac{2}{\sqrt{m}}V_f^s + \frac{2}{\sqrt{m}}V_f + \frac{1}{m}V_f\right), \qquad F := \frac{2\,\tilde{V}_f\,A}{B},$$

*with the understanding that if $B = 0$ then the bound below reduces to $(C' - 1) \cdot 4\tilde{V}_f$ (achieved in the limit $a \to \infty$). Define $y^*$ to be the unique positive real root of $y^3 - 8Fy - 16FA = 0$, namely*

$$A^2 \geq \frac{8F}{27} \implies y^* = \sqrt[3]{8F\left(A + \sqrt{A^2 - \tfrac{8F}{27}}\right)} + \sqrt[3]{8F\left(A - \sqrt{A^2 - \tfrac{8F}{27}}\right)},$$

$$A^2 < \frac{8F}{27} \implies y^* = 4\sqrt{\frac{2F}{3}}\,\cos\left(\frac{1}{3}\arccos\left(3A\sqrt{\frac{3}{8F}}\right)\right).$$

*Let $a^\star := 2A + y^*$ and define the constrained optimizer $a_{\text{opt}} := \max\{5, a^\star\}$. Then*

$$\text{err}^{\text{NCC}}_{m,D}(f) \leq (C' - 1)\,E(a_{\text{opt}}), \qquad E(a) = \left(\tfrac{1}{2} - \tfrac{2}{a} - \tfrac{2^{3/2}}{am}\right)^{-2}\tilde{V}_f + B\,a.$$

*Proof.* Starting from

$$\text{err}^{\text{NCC}}_{m,D}(f) \leq (C' - 1)\inf_{a \geq 5}\left[\left(\tfrac{1}{2} - \tfrac{2}{a} - \tfrac{2^{3/2}}{am}\right)^{-2}\tilde{V}_f + \frac{a}{4}\left(\tfrac{2}{\sqrt{m}}V_f^s + \tfrac{2}{\sqrt{m}}V_f + \tfrac{1}{m}V_f\right)\right],$$

write $E(a) = c(a)^{-2}\tilde{V}_f + Ba$ with $c(a) := \tfrac{1}{2} - \tfrac{A}{a} = \tfrac{a - 2A}{2a}$, $A = 2 + \tfrac{2^{3/2}}{m}$, and $B$ as above. On $(2A, \infty)$ one has

$$E'(a) = -\frac{2\tilde{V}_f A}{a^2 c(a)^3} + B, \qquad E''(a) = \frac{32A\tilde{V}_f(A + a)}{(2A - a)^4} > 0,$$

so $E$ is strictly convex with a unique stationary point $a^\star > 2A$ solving

$$a^2 c(a)^3 = \frac{2\tilde{V}_f A}{B} =: F.$$

Let $y = a - 2A$. Since $a^2 c(a)^3 = \dfrac{(a-2A)^3}{8a} = \dfrac{y^3}{8(y+2A)}$, the stationarity condition is $y^3 - 8Fy - 16FA = 0$. Cardano's formula gives the stated $y^* > 0$ and $a^\star = 2A + y^*$. At $a^\star$,

$$E(a^\star) = \frac{B}{4A}\, a^\star\big(a^\star + 2A\big)$$

by substituting the stationarity equation into $E$. Because $m \geq 10$ we have $2A < 5$, so the constrained minimizer on $[5, \infty)$ is $a_{\mathrm{opt}} = \max\{5, a^\star\}$ and $\inf_{a \geq 5} E(a) = E(a_{\mathrm{opt}})$, which yields the claim and its two cases. If $B = 0$, then $E(a) = c(a)^{-2}\tilde{V}_f$ decreases to $4\tilde{V}_f$ as $a \to \infty$, giving the stated simplification. $\qquad\square$

## D  Proof of Thm. 2

**Theorem 2.** *Let $d \geq C - 1$ and let $S = \{(x_i, y_i)\}_{i=1}^N \subset \mathcal{X} \times [C]$ be a balanced labeled dataset with $C$ classes. Suppose $f$ is a global minimizer of the supervised contrastive loss $\mathcal{L}^{\mathrm{NSCL}}(f)$ (over all functions $f : \mathcal{X} \to \mathbb{R}^d$). Then, the representations satisfy the following properties:*

1. ***Augmentation Collapse:*** *For each $i \in [N]$ and for every pair $l_1, l_2 \in [K]$, we have $z_i^{l_1} = z_i^{l_2}$.*

2. ***Within-Class Collapse:*** *For any two samples $x_i$ and $x_j$ with the same label ($y_i = y_j$), their representations coincide: $z_i = z_j$. Namely, each class has a unique class embedding.*

3. ***Simplex Equiangular Tight Frame:*** *Let $\{\mu_1, \ldots, \mu_C\}$ denote the set of class-center embeddings. These vectors form a simplex ETF in $\mathbb{R}^d$; specifically, they satisfy $\sum_{c=1}^C \mu_c = 0$, $\|\mu_c\|_2 = \|\mu_{c'}\|_2$ and $\langle \mu_c, \mu_{c'} \rangle = -\frac{\|\mu_c\|_2^2}{C-1}$ for all $c \neq c' \in [C]$.*

*Proof.* We divide the proof into three parts. In the first part, we restate and relax the problem. In the second part, we show that the embedding vectors of augmentations of the same sample collapse to the same vector. In the third part, we demonstrate that the embeddings of samples from the same class collapse to the same vectors, and the class vectors form a Simplex ETF.

**Restating the problem.**  Since we focus on the unconstrained features model [78, 79], we can rewrite $\mathcal{L}^{\mathrm{NSCL}}(f)$ in the following way:

$$\mathcal{L}^{\mathrm{NSCL}}(Z) = -\frac{1}{K^2 N} \sum_{l_1, l_2 = 1}^K \sum_{i=1}^N \log\left(\frac{\exp\big(\mathrm{sim}(z_i^{l_1}, z_i^{l_2})\big)}{\sum_{l_3=1}^K \sum_{\substack{j=1 \\ y_j \neq y_i}}^N \exp\big(\mathrm{sim}(z_i^{l_1}, z_j^{l_3})\big)}\right),$$

where $Z = (z_i^l)_{i \in [N], l \in [K]}$ denotes a collection of learnable representations, where each $z_i^l$ is the embedding of sample $x_i$ under augmentation $l$.

Since one may either normalize the vectors externally or inside the cosine similarity and because unit norm constraint is stronger than a norm upper bound constraint, we have:

$$\min_Z \left\{ -\frac{1}{K^2 N} \sum_{l_1, l_2 = 1}^K \sum_{i=1}^N \log\left(\frac{\exp\big(\mathrm{sim}(z_i^{l_1}, z_i^{l_2})\big)}{\sum_{l_3=1}^K \sum_{j: y_j \neq y_i}^N \exp\big(\mathrm{sim}(z_i^{l_1}, z_j^{l_3})\big)}\right) \right\}$$

$$= \min_Z \left\{ -\frac{1}{K^2 N} \sum_{l_1, l_2 = 1}^K \sum_{i=1}^N \log\left(\frac{\exp\big(\langle z_i^{l_1}, z_i^{l_2}\rangle\big)}{\sum_{l_3=1}^K \sum_{j: y_j \neq y_i}^N \exp\big(\langle z_i^{l_1}, z_j^{l_3}\rangle\big)}\right) \;\text{s.t.}\; \forall i \in [N], l \in [K]: \|z_i^l\|_2 = 1 \right\}$$

$$\geq \min_Z \left\{ \underbrace{-\frac{1}{K^2 N} \sum_{l_1, l_2 = 1}^K \sum_{i=1}^N \log\left(\frac{\exp\big(\langle z_i^{l_1}, z_i^{l_2}\rangle\big)}{\sum_{l_3=1}^K \sum_{j: y_j \neq y_i}^N \exp\big(\langle z_i^{l_1}, z_j^{l_3}\rangle\big)}\right)}_{=:\mathcal{Q}(Z)} \;\text{s.t.}\; \forall i \in [N], l \in [K]: \|z_i^l\|_2 \leq 1 \right\}.$$

We will show that the global minimum of the last optimization problem is obtained when we have augmentation collapse, within-class collapse, and a simplex ETF behavior. In particular, this will give us the result that the solution to the latter problems is achieved in the same way.

**Augmentation Collapse.** We begin by proving that for every $i_0 \in [N]$, the vectors $z_{i_0}^1, \ldots, z_{i_0}^K$ are identical at any global minimum. Fix some index $i_0 \in [N]$ and suppose for contradiction that there exist two distinct augmentations $l_1^* \neq l_2^*$ with $z_{i_0}^{l_1^*} \neq z_{i_0}^{l_2^*}$. Define the averaged vector $\tilde{z}_{i_0} = \frac{1}{K} \sum_l z_{i_0}^l$ and let $\tilde{Z}$ be the collection obtained by replacing $z_{i_0}^l$ by $\tilde{z}_{i_0}$ for all $l \in [K]$. We will show that $\mathcal{Q}(Z) > \mathcal{Q}(\tilde{Z})$.

Consider the loss function:

$$\mathcal{Q}(Z) = -\frac{1}{K^2 N} \sum_{i=1}^{N} \sum_{l_1, l_2 = 1}^{K} \log \left( \frac{\exp\left(\langle z_i^{l_1}, z_i^{l_2} \rangle\right)}{\sum_{l_3=1}^{K} \sum_{j: y_j \neq y_i} \exp\left(\langle z_i^{l_1}, z_j^{l_3} \rangle\right)} \right).$$

Now, fix the index $i_0$ and split the sum over $i$ into the contribution from $i_0$ and the contributions from all other indices $i \neq i_0$. In addition, let $\delta_{i,j}^{l_1, l_2} = \exp\left(\langle z_i^{l_1}, z_j^{l_2} \rangle\right)$. With this separation, we obtain

$$
\begin{aligned}
\mathcal{Q}(Z) \;=\; & -\frac{1}{K^2 N} \sum_{l_1, l_2 = 1}^{K} \sum_{i=1}^{N} \log \left( \frac{\exp\left(\langle z_i^{l_1}, z_i^{l_2} \rangle\right)}{\sum_{l_3=1}^{K} \sum_{j: y_j \neq y_i} \exp\left(\langle z_i^{l_1}, z_j^{l_3} \rangle\right)} \right) \\
=\; & \frac{1}{N} \left[ -\frac{1}{K^2} \sum_{l_1, l_2 = 1}^{K} \langle z_{i_0}^{l_1}, z_{i_0}^{l_2} \rangle - \frac{1}{K^2} \sum_{l_1, l_2 = 1}^{K} \sum_{i \neq i_0} -\langle z_i^{l_1}, z_i^{l_2} \rangle \right] \\
& + \frac{1}{K^2 N} \sum_{l_1, l_2 = 1}^{K} \sum_{i \neq i_0}^{N} \log \left( \sum_{l_3=1}^{K} \sum_{j: y_j \neq y_i} \exp\left(\langle z_i^{l_1}, z_j^{l_3} \rangle\right) \right) \\
& + \frac{1}{KN} \sum_{l_1=1}^{K} \log \left( \sum_{l_3=1}^{K} \sum_{j: y_j \neq y_{i_0}} \exp\left(\langle z_{i_0}^{l_1}, z_j^{l_3} \rangle\right) \right).
\end{aligned}
$$

Similarly,

$$
\begin{aligned}
\mathcal{Q}(\tilde{Z}) \;=\; & \frac{1}{N} \left[ -\langle \tilde{z}_{i_0}, \tilde{z}_{i_0} \rangle - \frac{1}{K^2} \sum_{l_1, l_2 = 1}^{K} \sum_{i \neq i_0} -\langle z_i^{l_1}, z_i^{l_2} \rangle \right] \\
& + \frac{1}{K^2 N} \sum_{l_1, l_2 = 1}^{K} \sum_{i \neq i_0}^{N} \log \left( \sum_{l_3=1}^{K} \sum_{\substack{j \in [N] \\ y_j \neq y_i \\ j \neq i_0}} \exp\left(\langle z_i^{l_1}, z_j^{l_3} \rangle\right) + K \exp\left(\langle z_i^{l_1}, \tilde{z}_{i_0} \rangle\right) \right) \\
& + \frac{1}{N} \log \left( \sum_{l_3=1}^{K} \sum_{j: y_j \neq y_{i_0}} \exp\left(\langle \tilde{z}_{i_0}, z_j^{l_3} \rangle\right) \right).
\end{aligned}
$$

By definition, since $\tilde{z}_{i_0} = \frac{1}{K} \sum_l z_{i_0}^l$, its squared norm is given by

$$\langle \tilde{z}_{i_0}, \tilde{z}_{i_0} \rangle \;=\; \|\tilde{z}_{i_0}\|_2^2 \;=\; \frac{1}{K^2} \sum_{l_1, l_2 = 1}^{K} \langle z_{i_0}^{l_1}, z_{i_0}^{l_2} \rangle.$$

Since $z_i^{l_1} \neq 0$ for every $i \in [N]$ and $l_1 \in [K]$ (otherwise the cosine similarity would be undefined), the function

$$h_i^{l_1}(x) \;=\; \exp\left(\langle z_i^{l_1}, x \rangle\right)$$

is convex in $x$. Consequently, by Jensen's inequality, for any subset $S$ and any collection of vectors $\{z_j^{l_3} : j \in S\}$ we have

$$\frac{1}{|S|} \sum_{j \in S} \exp\left(\langle z_i^{l_1}, z_j^{l_3} \rangle\right) \;\geq\; \exp \left( \left\langle z_i^{l_1}, \frac{1}{|S|} \sum_{j \in S} z_j^{l_3} \right\rangle \right),$$

with equality if and only if all vectors $z_j^{l_3}$ in the subset are identical.

Applying this inequality to the relevant subsets yields

$$\log \left( \sum_{l_3=1}^{K} \sum_{j: y_j \neq y_i} \exp\left(\langle z_i^{l_1}, z_j^{l_3} \rangle\right) \right) > \log \left( \sum_{l_3=1}^{K} \sum_{\substack{j: y_j \neq y_i \\ j \neq i_0}} \exp\left(\langle z_i^{l_1}, z_j^{l_3} \rangle\right) + K \exp\left(\langle z_i^{l_1}, \tilde{z}_{i_0} \rangle\right) \right),$$

where the strict inequality follows from the assumption that for some indices $l_1^*$ and $l_2^*$ (with $l_1^* \neq l_2^*$), we have $z_{i_0}^{l_1^*} \neq z_{i_0}^{l_2^*}$.

Next, define the function

$$G(x) = \log\left( \sum_{l_3=1}^{K} \sum_{j : y_j \neq y_{i_0}} \exp(\langle x, z_j^{l_3} \rangle) \right).$$

It is well known that $G(x)$ is convex in $x$. Therefore, by Jensen's inequality we obtain

$$\frac{1}{K} \sum_{l_1=1}^{K} G(z_{i_0}^{l_1}) \geq G\left( \frac{1}{K} \sum_{l_1=1}^{K} z_{i_0}^{l_1} \right),$$

which can be written equivalently as

$$\frac{1}{K} \sum_{l_1=1}^{K} \log\left( \sum_{l_3=1}^{K} \sum_{j : y_j \neq y_{i_0}} \exp(\langle z_{i_0}^{l_1}, z_j^{l_3} \rangle) \right) \geq \log\left( \sum_{l_3=1}^{K} \sum_{j : y_j \neq y_{i_0}} \exp(\langle \tilde{z}_{i_0}, z_j^{l_3} \rangle) \right).$$

Combining this inequality with the previous expansion, we deduce that $\mathcal{Q}(\tilde{Z}) < \mathcal{Q}(Z)$, which contradicts the assumption that $Z$ is a global minimizer of the loss. Consequently, for each $i_0 \in [N]$, it must be that all vectors $z_{i_0}^1, z_{i_0}^2, \ldots, z_{i_0}^K$ are equal at a global minimum.

**Within-class collapse and Simplex ETF.** Since the augmentations collapse, with no loss of generality we assume that $K = 1$. We now analyze the embeddings $z_i$ by applying the arithmetic–geometric mean (AGM) inequality to the $n(C-1)$ positive numbers

$$\left\{ \exp(\langle z_i, z_j \rangle) \mid j : y_j \neq y_i \right\},$$

for each fixed $i \in [N]$. The AGM inequality yields

$$\frac{1}{n(C-1)} \sum_{j : y_j \neq y_i} \exp(\langle z_i, z_j \rangle) \geq \left( \prod_{j : y_j \neq y_i} \exp(\langle z_i, z_j \rangle) \right)^{\frac{1}{n(C-1)}}, \tag{14}$$

Taking natural logarithms on both sides gives

$$\log\left( \sum_{j : y_j \neq y_i} \exp(\langle z_i, z_j \rangle) \right) \geq \log(n(C-1)) + \frac{1}{n(C-1)} \sum_{j : y_j \neq y_i} \langle z_i, z_j \rangle.$$

Averaging this inequality over $i \in [N]$ yields

$$\frac{1}{N} \sum_{i=1}^{N} \log\left( \sum_{j : y_j \neq y_i} \exp(\langle z_i, z_j \rangle) \right) \geq \log(n(C-1)) + \frac{1}{Nn(C-1)} \sum_{i=1}^{N} \sum_{j : y_j \neq y_i} \langle z_i, z_j \rangle. \tag{15}$$

Notice that the double sum of inner products can be rewritten as

$$\begin{aligned}
\sum_{i=1}^{N} \sum_{j : y_j \neq y_i} \langle z_i, z_j \rangle &= \sum_{i,j=1}^{N} \langle z_i, z_j \rangle - \sum_{i=1}^{N} \sum_{j : y_j = y_i} \langle z_i, z_j \rangle \\
&= \left\| \sum_{i=1}^{N} z_i \right\|_2^2 - \sum_{c=1}^{C} \left\| \sum_{i : y_i = c} z_i \right\|_2^2 \\
&= \left\| n \sum_{c=1}^{C} \mu_c \right\|_2^2 - \sum_{c=1}^{C} \| n \mu_c \|_2^2
\end{aligned}$$

where we define the class-mean embeddings as $\mu_c = \frac{1}{n} \sum_{i : y_i = c} z_i$. Substituting this identity into (15) yields

$$\frac{1}{N} \sum_{i=1}^{N} \log\left( \sum_{j : y_j \neq y_i} \exp(\langle z_i, z_j \rangle) \right) \geq \log(n(C-1)) + \frac{1}{C(C-1)} \left( \left\| \sum_{c=1}^{C} \mu_c \right\|_2^2 - \sum_{c=1}^{C} \| \mu_c \|_2^2 \right). \tag{16}$$

By Jensen's inequality,

$$\|\mu_c\|_2^2 = \left\|\sum_{i:y_i=c} z_i\right\|_2^2 \leq \tfrac{1}{n}\sum_{i:y_i=c}\|z_i\|_2^2 \leq 1, \tag{17}$$

it follows that

$$\sum_{i=1}^{N}\sum_{j:y_j\neq y_i}\langle z_i,z_j\rangle \geq n^2\left\|\sum_{c=1}^{C}\mu_c\right\|_2^2 - n^2C \tag{18}$$

Substituting this estimate into (15) gives

$$\frac{1}{N}\sum_{i=1}^{N}\log\left(\sum_{j:y_j\neq y_i}\exp(\langle z_i,z_j\rangle)\right) \geq \log(n(C-1)) + \frac{1}{C(C-1)}\left(\left\|\sum_{c=1}^{C}\mu_c\right\|_2^2 - C\right).$$

Substituting the bound (16) into the expression for $\mathcal{Q}(Z)$ (the loss) results in

$$\mathcal{Q}(Z) \geq -\frac{1}{N}\sum_{i=1}^{N}\|z_i\|_2^2 + \log(n(C-1)) + \frac{1}{C(C-1)}\left(\left\|\sum_{c=1}^{C}\mu_c\right\|_2^2 - C\right). \tag{19}$$

Since

$$-\frac{1}{N}\sum_{i=1}^{N}\|z_i\|_2^2 \geq -1 \quad\text{and}\quad \left\|\sum_{c=1}^{C}\mu_c\right\|_2^2 \geq 0, \tag{20}$$

we obtain the lower bound

$$\mathcal{Q}(Z) \geq \log\big(n(C-1)\big) - 1 - \frac{1}{C-1}. \tag{21}$$

Equality in (21) is achieved only if all the preceding inequalities hold as equalities. In particular, equality in (20) requires $\sum_{c=1}^{C}\mu_c = 0$ and $\forall i \in [N] : \|z_i\|_2 = 1$. Similarly, equality in (17) (used in the derivation) forces $\left\|\sum_{i:y_i=c} z_i\right\|_2^2 = \tfrac{1}{n}\sum_{i:y_i=c}\|z_i\|_2^2$ which, via Jensen's inequality, implies that $z_i = \mu_{y_i}$ for all $i \in [N]$. Furthermore, equality in (14) is attained if and only if

$$\forall i : \exp(\langle z_i,z_j\rangle) \text{ is constant w.r.t. } j \text{ such that } y_j \neq y_i.$$

In particular, by applying natural logarithm in both sides, and considering that fact that $z_i = \mu_{y_i}$

$$\forall c \in [C] : \langle \mu_c,\mu_{c'}\rangle \text{ is constant for all } c' \neq c.$$

In particular, $\langle \mu_i,\mu_j\rangle = \langle \mu_r,\mu_j\rangle = \langle \mu_r,\mu_t\rangle$ and we have the more general equation:

$$\langle \mu_c,\mu_{c'}\rangle \text{ is constant for all } c \neq c' \in [C].$$

Together, these conditions exactly characterize a simplex equiangular tight frame (ETF); in particular, they imply that

$$\|\mu_c\|_2 = 1, \quad \sum_{c=1}^{C}\mu_c = 0, \quad\text{and}\quad \langle \mu_c,\mu_{c'}\rangle = -\frac{1}{C-1} \text{ forall } c \neq c' \in [C].$$

A straightforward calculation shows that

$$\mathcal{Q}(Z) = -1 + \frac{1}{C}\sum_{i=1}^{C}\log\left((C-1)\exp\left(-\tfrac{1}{C-1}\right)\right) = \log(C-1) - 1 - \frac{1}{C-1},$$

so that the lower bound in (19) is attained. Consequently, a set of vectors $Z$ is a global minimizer of $\mathcal{Q}(Z)$ if and only if it forms a simplex equiangular tight frame. $\qquad\square$

