# OpenReview forum: "Self-Supervised Contrastive Learning is Approximately Supervised Contrastive Learning"
_NeurIPS.cc/2025/Conference — NeurIPS 2025 poster_

### Official Review · Reviewer_J3ay · 2025-06-25

**Clarity:** 4
**Significance:** 3
**Originality:** 3
**Rating:** 5
**Confidence:** 3

**Summary:**

This paper studies a duality between contrastive learning (CL, self-supervised) and negatives-only supervised contrastive learning (NSCL, where samples belonging to the same class as the positive sample are removed from the denominator).
First, it shows that the standard contrastive learning approximates the NSCL and that the gap between loss values of CL and NSCL decreases with the number of classes with $\frac{1}{C-1}$ (for a balanced dataset).
Next, it introduces a new bound on few-shot error of linear probing. Compared to previous work, this bound depends not only on class-distance-normalized variance (CDNV) between classes, but also on its *directinal* variant, i.e., on *directional* CDNV, which measures the variation along the line between centers of the respective classes. Moreover, the authors show that this newly introduced directional CDNV dominates the bound on few-shot error of linear probing.
Finally, the findings presented in this paper are empirically validated on four datasets (CIFAR10, CIFAR100, mini-ImageNet, SVHN) using SimCLR or MoCov2 algorithm (with DCL loss).

**Questions:**

- How do you define the "same point" in the augmentation collapse on L66? Is it some small neighborhood?
- What does "upper bound" in Fig. 2 (top) represent?

**Ethical Concerns:**

["NO or VERY MINOR ethics concerns only"]

**Final Justification:**

I've read the rebuttal and appreciate the insights and new experiments you presented. It helped me to understand the paper better and answered my questions. I see it important to include these in the final version of the paper, also with the discussion of why the gap in Figure 3 slightly increases (this was a shared question with another reviewer).

I maintain my original rating of 5 - Accept.

**Limitations:**

yes

**Quality:**

4

**Strengths And Weaknesses:**

**Strengths**
- The paper is well written with a clear structure and story.
- The findings reported in this paper are valuable in the steps towards a better theoretical understanding of contrastive learning.
- Good visualizations and plots that support the findings.
- The findings are well documented and commented on. The notation of the equations is clear.
- The reported contributions are empirically validated on four datasets using either SimCLR or MoCov2 algorithm with a nice level of detail given about the experiments.

**Weaknesses**
- The closseness of the two loss values (for DCL and NSCL) does not directly imply the same behavior and structure of the learned representation. I am grateful that this is also commented on by the authors.
- (minor) Sometimes, the generally-used variables (e.g., *K* on L220) are introduced earlier in the text and it can be a bit confusing.
- Experiments on a larger dataset (e.g., ImageNet) are missing.
- It is not explained what "upper bound" in Fig 2 (top) stands for?
- Claims on L298-299 are not discussed. What can be the reason for the gap between the losses to increase?

---

> ### Author Rebuttal · Authors · 2025-07-31
>
> **Note:** We thank the reviewer for their very thoughtful feedback and comments. Following the reviewers’ suggestions, we conducted multiple experiments to strengthen our work. Since there is no dedicated common space for experiments, we **present them at the beginning of the responses to reviewers 8fbK (part 1) and J3ay (part 2)**.
>
> ---
> ---
>
> ## Summary of experiments (part 2)
>
> **Experiment (5)** We reproduced the experiment in Fig. 5 with the **ViT architecture** on **CIFAR10, CIFAR100, mini-ImageNet and Tiny-ImageNet**.
>
> ### *CIFAR10*
>
>
> |       **Epochs >**            | **0**   | **10**  | **100** | **1000** | **2000** |
> |-------------------|---------|---------|---------|----------|----------|
> | **DCL CDNV**      | 11.39   | 2.90    | 1.67    | 1.42     | 2.00     |
> | **NSCL CDNV**     | 10.31   | 2.38    | 0.43    | 0.06     | 0.05     |
> | **DCL Dir‑CDNV**  | 1.88    | 0.27    | 0.10    | 0.05     | 0.05     |
> | **NSCL Dir‑CDNV** | 1.99    | 0.25    | 0.05    | 0.02     | 0.02     |
>
> ### *CIFAR100*
>
>
> |       **Epochs >**            | **0**   | **10**  | **100** | **1000** | **2000** |
> |-------------------|---------|---------|---------|----------|----------|
> | **DCL CDNV**      | 6.90    | 3.85    | 3.35    | 3.015    | 3.25     |
> | **NSCL CDNV**     | 6.14    | 3.91    | 2.73    | 1.24     | 0.83     |
> | **DCL Dir‑CDNV**  | 1.23    | 0.14    | 0.07    | 0.06     | 0.06     |
> | **NSCL Dir‑CDNV** | 1.01    | 0.15    | 0.07    | 0.06     | 0.05     |
>
> ### *Mini-imagenet*
>
>
> |       **Epochs >**            | **0**   | **10**  | **100** | **1000** | **2000** |
> |-------------------|---------|---------|---------|----------|----------|
> | **DCL CDNV**      | 5.78    | 2.75    | 2.51    | 1.69     | 1.63     |
> | **NSCL CDNV**     | 6.12    | 2.73    | 2.34    | 1.06     | 0.64     |
> | **DCL Dir‑CDNV**  | 1.03    | 0.20    | 0.09    | 0.05     | 0.04     |
> | **NSCL Dir‑CDNV** | 1.04    | 0.20    | 0.09    | 0.04     | 0.04     |
>
> ### *Tiny-Imagenet*
>
> | **Epochs >** | **0** | **10** | **100** | **500** |
> | ----------------- | ----- | ------ | ------- | ------- |
> | **DCL CDNV** | 6.49 | 4.06 | 3.69 | 3.08 |
> | **NSCL CDNV** | 6.29 | 4.13 | 3.47 | 2.58 |
> | **DCL Dir-CDNV** | 1.03 | 0.14 | 0.08 | 0.06 |
> | **NSCL Dir-CDNV** | 0.87 | 0.14 | 0.08 | 0.06 |
>
> **Experiment (6)** We reproduced the experiment in Fig. 2 (top) with a ResNet-50 trained with the CL loss instead of the DCL loss:
> ### **CIFAR10**
> | Epochs | CL     | DCL    | NSCL   | Upper Bound |
> |--------|--------|--------|--------|-------------|
> | 0      | 8.2898 | 8.2895 | 8.1843 | 8.7837      |
> | 10     | 7.8307 | 7.8303 | 7.7079 | 8.3073      |
> | 100    | 7.6074 | 7.6068 | 7.4649 | 8.0643      |
> | 1000   | 7.5065 | 7.5060 | 7.3545 | 7.9539      |
> ### **Mini-imagenet (C=100)**
> | Epochs | CL     | DCL    | NSCL   | Upper Bound |
> |--------|--------|--------|--------|-------------|
> | 0      | 7.3899 | 7.3892 | 7.3794 | 7.4515      |
> | 10     | 6.7831 | 6.7819 | 6.7703 | 6.8422      |
> | 100    | 6.5832 | 6.5817 | 6.5693 | 6.6413      |
> | 1000   | 6.5592 | 6.5578 | 6.5451 | 6.6171      |
>
> **Experiment (7)** We trained pairs of CL or DCL and NSCL-trained models starting from the same initialization, trained using the same batches at each iteration. In order to evaluate the similarity/alignment between the pairs of models we used the Centered Kernel Alignment (CKA) (https://proceedings.mlr.press/v97/kornblith19a/kornblith19a.pdf) and Representation Similarity Analysis (RSA) (https://www.frontiersin.org/journals/systems-neuroscience/articles/10.3389/neuro.06.004.2008/full) metrics (0 = no alignment, 1 = perfect alignment). As can be seen, these paired models achieved very high CKA and RSA values, demonstrating that the learned representations are strongly aligned across the two training objectives.
>
> |                | **CIFAR10** |        | **CIFAR100** |        | **tiny-ImageNet** |        |
> |----------------|-------------|--------|--------------|--------|-------------------|--------|
> |                | **RSA**     | **CKA**| **RSA**      | **CKA**| **RSA**           | **CKA**|
> | **ViT (DCL)**  | 0.8599        | 0.8202   | 0.9073       | 0.9099 | 0.8918            | 0.8945 |
> | **ResNet50 (CL)** | 0.8338   | 0.8102 | 0.9104         | 0.9113  | 0.8864            | 0.8974 |
>
> **Experiment (8)** We verified augmentation collapse by training DCL ViT models and measuring cosine similarity (0 = no collapse, 1 = perfect collapse). We measured the averaged cosine similarity between the embeddings of augmentations of the same sample and the cosine similarity between augmentations of different samples (over the full dataset and 5 augmentation for each pair of samples). Same‑sample augmentations averaged ~0.90, while different‑sample augmentations averaged < 0.15, confirming highly aligned embeddings of augmentations of the same sample.
>
> | **CIFAR10 (same)** | **CIFAR10 (different)** | **CIFAR100 (same)** | **CIFAR100 (different)** | **mini-ImageNet (same)** | **mini-ImageNet (different)** |
> |------------------|-----------------------|-------------------|------------------------|------------------------|---------------------------|
> | 0.9155             | 0.1321                  | 0.9091              | 0.1077                   | 0.8875                   | 0.0650                     |
>
> ---
> ---
>
> ## Response to J3ay
>
> > **Reviewer:** The closseness of the two loss values (for DCL and NSCL) does not directly imply the same behavior and structure of the learned representation. I am grateful that this is also commented on by the authors.
>
> You are absolutely right: a small difference in objective values does not by itself guarantee identical embedding geometry. The loss‑gap bound is only the first step. To better understand representation‑level similarity we: (i) Introduce directional CDNV (Sec. 3.2): a geometry‑aware variance measure that both DCL and NSCL implicitly drive to low values during training and (ii) link directional CDNV to downstream accuracy (Prop. 1, Sec. 3.2).
>
> To further substantiate the alignment between CL/DCL‑ and NSCL‑trained models, we ran dedicated experiments quantifying representation similarity. We trained matched CL/DCL and NSCL models from the same initialization with identical batches at each iteration and measured alignment using Centered Kernel Alignment (CKA) and Representational Similarity Analysis (RSA)—both standard embedding‑similarity metrics (ranging between 0–1). As shown in experiment (7) (in the response to J3ay), the paired models achieve very high CKA and RSA values, indicating that their learned representations are strongly aligned.
>
> > **Reviewer:** (minor) Sometimes, the generally-used variables (e.g., K on L220) are introduced earlier in the text and it can be a bit confusing.
>
> Thanks for pointing this out. We will revisit these widely used variables later in the text as a reminder.
>
> > **Reviewer:** Experiments on a larger dataset (e.g., ImageNet) are missing.
>
> Thank you for the suggestion. To scale-up our experiments, we conducted plenty of experiments with the ViT architecture, as well as the ImageNet-1k (1000 classes) and Tiny-ImageNet (200 classes) datasets. The experiments reproduce Fig. 2, 4 and 5 (see experiments (1)–(5) in the response to J3ay).
>
> > **Reviewer:** It is not explained what "upper bound" in Fig 2 (top) stands for?
>
> The "upper bound" in Figure 2 refers to $_{\text{NSCL}} + \log(1 + e^2/(C − 1))$, as noted in the caption. We will clarify this further in the next version of the paper.
>
> > **Reviewer:** Claims on L298-299 are not discussed. What can be the reason for the gap between the losses to increase?
>
> We can express the loss difference as
> $$
> L_{\text{DCL}} - L_{\text{NSCL}} = \log\left(1 + \frac{A}{B}\right),
> $$
> where
> * $A = \sum_{j \neq i, y_j = y_i} e^{\text{sim}(z_i, z_j)}$ is the sum over same‑class negatives, and
> * $B = \sum_{y_j \neq y_i} e^{\text{sim}(z_i, z_j)}$ is the sum over all cross‑class negatives.
>
> Early in training, similarities between all samples (same or different class) are roughly equal, so $A/B$ is small—dominated by the fact that there are far fewer same‑class terms than cross‑class ones. As a result, the loss gap is negligible.
>
> As training progresses, same‑class samples become more similar while cross‑class similarities decrease. This increases $A$ relative to $B$, so the gap $\log(1 + A/B)$ grows slightly. The increase remains small because the same‑class terms are still a small fraction of the total. This explains the gradual upward trend observed in Fig. 3.
>
> The gap is still small because the same‑class negatives make up only a tiny fraction of all negatives, and the label‑agnostic bound keeps the difference tightly controlled.
>
> > **Reviewer:** How do you define the "same point" in the augmentation collapse on L66? Is it some small neighborhood?
>
> By "same point," we mean that for two augmentations $x'$ and $x''$ of the same sample, the embeddings satisfy $f(x')=f(x'')$; i.e., they are exactly equal. This definition arises from our analysis of global minimizers under the unconstrained features model, which is an idealized abstraction of the learning setting.
>
> In practice, exact equality is not achievable. Instead, augmentations of the same sample tend to produce very similar embeddings. Empirically, we observe high cosine similarity between such pairs: in experiment (8) (in the response to J3ay) the average cosine similarity between augmentations of the same sample was ~0.9, compared to <0.15 for augmentations of different samples (averaged over 5 augmentations for each pair of samples).

---

> > ### Comment · Reviewer_J3ay · 2025-08-05
> > **response to the rebuttal**
> >
> > Dear authors, thank you very much for providing the rebuttal. I appreciate the insights and new experiments you presented. It helped me to understand the paper better and answered my questions. I see it important to include these in the final version of the paper, also with the discussion of why the gap in Figure 3 slightly increases (this was a shared question with another reviewer).
> >
> > I maintain my original rating of 5 - Accept.

---

> > > ### Author Response · Authors · 2025-08-06
> > >
> > > Thank you very much for your constructive feedback and for reviewing our response. We’re glad that our experiments and additional explanations addressed your questions. In the final version, we will revise the text to include a detailed discussion of Figure 3 and incorporate all new experiments in a clear and organized manner.
> > >
> > > Thank you once again for your continued support and appreciation of our work.

---

### Official Review · Reviewer_8nSm · 2025-06-29

**Clarity:** 2
**Significance:** 2
**Originality:** 3
**Rating:** 4
**Confidence:** 4

**Summary:**

The authors provide theoretical foundations for self-supervised contrastive learning (CL) by investigating the gap between CL loss and a supervised negatives-only supervised contrastive loss (NSCL). The loss gap is proved to vanish as the number of classes increases. The paper also discusses that the NSCL-trained representations achieve accurate linear probing performance because the perfectly clustered representations lead to zero CDNV and directional CDNV. Experiments on benchmark image datasets are conducted to validate the theoretical findings.

**Questions:**

1. Figure 3 shows that the gap slightly increases over training. Is there any explanation?
2. In Figure 5, the CDNV of DCL increases at the end of training. Is there any explanation?

**Ethical Concerns:**

["NO or VERY MINOR ethics concerns only"]

**Final Justification:**

The rebuttal solved my concerns and I have raised my recommendation score to 4. However, I cannot give a higher rating because this paper investigates the geometry of the representation learned by NSCL instead of CL or DCL, and despite the loss values approximating each other, the behaviors of their representations are not alike.

**Quality:**

3

**Strengths And Weaknesses:**

**Strength.**

1. The paper provides theoretical foundations for self-supervised contrastive learning from a new perspective.
2. The theoretical claims are validated through experiments.

**Weakness.**
1. There is a mismatch between the main theorem and its proof. Theorem 1 states that the bound is for DCL, but according to its proof, the bound should be for CL. Also, the claims in Sections 3 and 4 are for DCL, whereas in the abstract, the gap is claimed to be between standard CL and NSCL losses.
2. Several related theoretical works are not mentioned, e.g., https://openreview.net/forum?id=XDJwuEYHhme, https://openreview.net/forum?id=c5K7gcd2Rw, https://openreview.net/forum?id=cN9GkuBo4w. The authors should discuss the connections and differences compared with these papers.
3. The concept of directional CDNV needs intuitive explanations. Some examples could help.
4. The theoretical claims show that DCL approximates NSCL when the number of classes is large, and that NSCL leads to perfectly clustered representations and accordingly good few-shot classification performance. However, the theoretical discussions fail to address why DCL leads to good representations. Specifically, do the theoretical claims mean that when the number of classes $C \to \infty$, the DCL representations are perfectly clustered? This doesn't seem to be right under my knowledge.
5. The experimental validation of diminishing loss gap is weak (Fig. 2). It only verifies $C=10$ and $C=100$. More results on Tinyimagenet (C=200) or Imagenet1k (C=1000) could further strengthen this claim.

---

> ### Author Rebuttal · Authors · 2025-07-31
>
> **Note:** We thank the reviewer for their very thoughtful feedback and comments. Following the reviewers’ suggestions, we conducted multiple experiments to strengthen our work. Since there is no dedicated common space for experiments, **we present them at the beginning of the responses to reviewers 8fbK (part 1) and J3ay (part 2)**.
>
> ---
> ---
>
> ## Response to 8nSm
>
> > **Reviewer:** There is a mismatch between the main theorem and its proof. Theorem 1 states that the bound is for DCL, but according to its proof, the bound should be for CL. Also, the claims in Sections 3 and 4 are for DCL, whereas in the abstract, the gap is claimed to be between standard CL and NSCL losses.
>
> 1. The proof actually covers both objectives; we stated Thm. 1 in terms of DCL only to keep notation light. The proof in fact covers both objectives, as shown by the chain of inequalities (that appear in the proof):
> $$
> \mathcal L_{\text{NSCL}} \le \mathcal L_{\text{DCL}} \le \mathcal L_{\text{CL}}
> \quad\text{and}\quad
> \mathcal L_{\text{CL}}-\mathcal L_{\text{NSCL}} \le \log\bigl(1+\tfrac{e^{2}}{C-1}\bigr),
> $$
> which yields
> $$
> 0 \le \mathcal L_{\text{DCL}}-\mathcal L_{\text{NSCL}}
> \le \mathcal L_{\text{CL}}-\mathcal L_{\text{NSCL}}
> \le \log\bigl(1+\tfrac{e^{2}}{C-1}\bigr),
> $$
> Thus the same bound holds for both CL and DCL.
>
> 2. We focused on DCL in the main text since it is an improved variant of the original CL objective (as reported in the original DCL paper). The abstract mentions standard CL because that term is more familiar and one can think of CL as a family of algorithms rather than necessarily a very specific implementation; we will revise it to read "(decoupled) contrastive loss" and add a sentence clarifying that Thm. 1 applies to both CL and DCL. We will also update the theorem statement and proof header to make this dual coverage more explicit.
>
> 3. Empirically, we conducted an experiment to reproduce Fig. 2 (top) for CL-training, showing the same behaviour as before (see experiment (6) in the response to J3ay). In addition, we evaluated CL, DCL and NSCL and the bound for DCL-trained models and observed negligible differences in the values of CL and DCL (see experiments (1) and (2) in the response to 8fbK).
>
> > **Reviewer:** Several related theoretical works are not mentioned, e.g., https://openreview.net/forum?id=XDJwuEYHhme, https://openreview.net/forum?id=c5K7gcd2Rw, https://openreview.net/forum?id=cN9GkuBo4w. The authors should discuss the connections and differences compared with these papers.
>
> We appreciate the reviewer highlighting these additional works. In Sec. 2 (lines 80–91) we already review 13 related SSL theory papers, but we are happy to add the suggested works and clarify their differences from ours.
>
> Both papers derive bounds on supervised terms (classification error in the first, supervised loss in the second), but just like the works mentioned in lines 80-91, their analyses rely on very strong assumptions that we avoid:
>
> (1) Thms. 1 and 6 in the first paper rely on Definition 1, the $(\varepsilon,\delta)$-augmentation condition. It effectively assumes that, for almost every pair of images in the same semantic class, there exists a sequence of the model’s augmentations that brings their augmented versions arbitrarily close. For a class such as “cat images”, that would mean crops, colour‑jitter, blurs, etc’, can morph one cat image into another, a claim that is clearly idealized and rarely satisfied in practice.
>
> (2) The bounds in the second paper (Thms. 2.1, 2.3, 2.4) depend on terms such as:
> $$
> \mathbb{E}\_{c}\mathbb{E}\_{\bar{x},\bar{x}’ \sim \rho\_{c}}\mathbb{E}\_{a}\min\_{a’}|| f(a(\bar{x})) - f(a’(\bar{x}’))||
> $$
> and
> $$
> \mathbb{E}\_{c}\mathbb{E}\_{\bar{x}’ \sim \rho\_{c}}\max\_{a,a’}|| f(a(\bar{x}’)) - f(a’(\bar{x}’))||
> $$
> which together essentially require embeddings of augmented samples from the same class to be close. While the second term can be somewhat controlled through optimization, the first term is far more restrictive: (i) it implicitly depends on class labels (which are unavailable to SSL algorithms) and (ii) it presupposes intra-class consistency across augmentations, a property that may not hold in practice. In effect, the analysis builds in the very property it seeks to prove (low intra-class variation in embeddings) in order to derive a bound on the supervised loss.
>
> > **Reviewer:** The concept of directional CDNV needs intuitive explanations. Some examples could help.
>
> Thank you for the suggestion. In order to illustrate the CDNV and directional CDNV (and to understand their difference), we created two plots for each one of those. The first plot includes two classes in 2‑D. We illustrate how CDNV measures the total spread of each class relative to the distance between their means—visualised as the radius of two equal‑radius "balls" around the class centres. For the directional CDNV, we keep the same denominator (inter‑centre distance) but now measure the spread only along the line that joins the two centres. In the figure this is visualized as the variance of the 1‑D projections of the samples onto that connecting line.
>
> This visual makes it clear why directional CDNV can be much smaller than CDNV—e.g., if the clouds are elongated orthogonally to the centre‑to‑centre axis, yet still dominates few‑shot error (Prop. 1). We will include these plots with an explanation in the final version.
>
> > **Reviewer:** The theoretical claims show that DCL approximates NSCL when the number of classes is large, and that NSCL leads to perfectly clustered representations and accordingly good few-shot classification performance. However, the theoretical discussions fail to address why DCL leads to good representations. Specifically, do the theoretical claims mean that when the number of classes $C \to \infty$, the DCL representations are perfectly clustered? This doesn't seem to be right under my knowledge.
>
> To clarify, we do not argue that DCL leads to perfectly clustered representations as $C \to \infty$. When $C \to \infty$ then we are able to perfectly minimize the NSCL loss (which leads to perfect clustering) only if the embedding dimension $d \to \infty$. Thus, in any realistic setting where $d$ is finite we will always have a noticeable gap between the two losses. Instead, we show that the DCL and NSCL losses are close which does not directly lead to the same level of clustering in the representations. In fact, similar to the proof of Thm. 2, it is possible to show that the global minimizer of DCL (which is only possible when the embedding dimension $d$ exceeds the number of samples $N$), the embeddings of training samples form an equiangular tight frame (ETF) of $N$ vectors. In that configuration, the directional‑CDNV is exactly zero (and the CDNV to be proportional to the number of samples per class), which is a weaker notion of clustering than the one given by minimizing the NSCL loss.
>
> > **Reviewer:** The experimental validation of diminishing loss gap is weak (Fig. 2). It only verifies $C=10$ and $C=100$. More results on Tinyimagenet (C=200) or Imagenet1k (C=1000) could further strengthen this claim.
>
> Following the reviews, we conducted additional experiments on both ImageNet-1k and TinyImageNet, as well as further experiments with the ViT architecture (see experiments (1)-(5) in the responses to 8fbK and J3ay).
>
> > **Reviewer:** Figure 3 shows that the gap slightly increases over training. Is there any explanation?
>
> We can express the loss difference as
> $$
> L_{\text{DCL}} - L_{\text{NSCL}} = \log\left(1 + \frac{A}{B}\right),
> $$
> where
> * $A = \sum_{j \neq i, y_j = y_i} e^{\text{sim}(z_i, z_j)}$ is the sum over same‑class negatives, and
> * $B = \sum_{y_j \neq y_i} e^{\text{sim}(z_i, z_j)}$ is the sum over all cross‑class negatives.
> Early in training, similarities between all samples (same or different class) are roughly equal, so $A/B$ is small—dominated by the fact that there are far fewer same‑class terms than cross‑class ones. As a result, the loss gap is negligible.
>
> As training progresses, same‑class samples become more similar while cross‑class similarities decrease. This increases $A$ relative to $B$, so the gap $\log(1 + A/B)$ grows slightly. The increase remains small because the same‑class terms are still a small fraction of the total. This explains the gradual upward trend observed in Fig. 3.
>
> > **Reviewer:** In Figure 5, the CDNV of DCL increases at the end of training. Is there any explanation?
>
> Thanks for bringing this up. Throughout the experiments we have consistently observed a phenomenon we refer to as "over‑training" in CL/DCL. At the first stage of training, the directional‑CDNV decays rapidly while the CDNV remains fairly stable (and may even decrease slightly). As training progresses, because the model continues to push samples apart (while still enforcing consistency within augmented views), the CDNV eventually starts to grow again, while the directional‑CDNV continues to decay (or saturates).
>
> Similar to the proof of Thm. 2, it is possible to show that if the embedding dimension $d$ exceeds the number of samples $N$ (an unrealistic but illustrative case), the global minimizer of CL/DCL organizes the embeddings of training samples as an equiangular tight frame (ETF) of $N$ vectors. In that configuration, the directional‑CDNV is exactly zero but the CDNV is proportional to the number of training samples per class.
>
> We will add discussion of this behavior and its origins in the next version of the paper.

---

> > ### Comment · Reviewer_8nSm · 2025-08-05
> >
> > I thank the authors for the rebuttal. I am happly to see that the proofs are rigor and the empirical gap between DCL and NSCL is supported by theoretical explanations. The explanations and additional experiments solved my concerns and I will raise my recommendation scores towards weak acceptance. However, I cannot give a higher rating because this paper investigates the geometry of the representation learned by NSCL instead of CL or DCL, and despite the loss values approximating each other, the behaviors of their representations are not alike. I also suggest that the authors clarify the use of CL or DCL throughout the paper in their next version, which could significantly improve readability and avoid misunderstandings.

---

> > > ### Author Response · Authors · 2025-08-06
> > >
> > > Thank you very much for your constructive feedback and for reviewing our response. We sincerely appreciate your decision to raise the recommendation score for our paper.
> > >
> > > We’re glad that our theoretical explanations and additional experiments helped address your concerns. We will incorporate both the explanations and the new experimental results into the final version of the paper. Thank you also for your suggestion to clarify the use of CL and DCL, we will revise the text accordingly to improve clarity and avoid potential misunderstandings.
> > >
> > > We truly appreciate your thoughtful comments and the opportunity to further improve our work.

---

### Official Review · Reviewer_WmN5 · 2025-07-02

**Clarity:** 2
**Significance:** 3
**Originality:** 3
**Rating:** 4
**Confidence:** 4

**Summary:**

In this paper, the author aims to establish a connection between contrastive learning and supervised learning. Through theoretical analysis, the author demonstrates how self-supervised contrastive learning can be mapped to a supervised contrastive loss that uses only negative samples. Both the upper and lower bounds of the proposed model are provided. Experiments involving few-shot classification show that as the number of classes increases, the behavior of the two paradigms becomes increasingly similar.

**Questions:**

Please refer to the weakness.

**Ethical Concerns:**

["NO or VERY MINOR ethics concerns only"]

**Final Justification:**

The rebuttal has addressed most of my concerns. The method proposed in this paper offers theoretical insights into contrastive learning and presents a potential framework for evaluating and guiding pretraining strategies. I believe these contributions can benefit the community and support the development of more effective self-supervised learning methods. Therefore, I consider this paper to be above the acceptance threshold.

**Limitations:**

yes

**Quality:**

3

**Strengths And Weaknesses:**

Strengths:

- The paper is well written and easy to follow.

- The two theorems proposed in the paper are mathematically well-formulated and rigorously proved.

- The experimental design effectively validates the theoretical claims made by the authors.

Weakness:
- The experiment conducted by the author is quite small in scale, as CIFAR-10, CIFAR-100, and mini-ImageNet each contain only 50,000 images. The results could be easily influenced by overfitting, especially in the presence of noisy samples. I am curious whether this phenomenon still holds when the number of samples per class increases or when the data becomes noisier. Nowadays, pretraining typically requires massive and often noisy datasets. Further experiments should be conducted to verify the author’s claims.

- The central claim that standard contrastive learning approximates a supervised variant under a growing number of classes is conceptually intuitive and has been suggested in prior works. While Theorem 1 formalizes this intuition with a clean bound, the assumption that the number of classes tends to infinity makes the result less surprising and arguably less practical. The paper ultimately confirms that contrastive objectives yield clustering behavior similar to supervised objectives, but this is already an implicit assumption in much of the literature. Thus, the theoretical contribution, though correct, feels incremental rather than insightful.

- This paper does not provide practical guidance for designing CL systems. For example, it does not address important  questions such as how augmentation affects learning, the robustness of  the theoretical results under incomplete augmentation, or whether CL can outperform supervised learning in all cases. Although the authors  introduce an upper bound on the few-shot error, it relies on quantities that are difficult to explicitly control during learning (e.g.,  class-wise variance) and has limited practical impact on accuracy.  Moreover, minimizing loss does not directly translate into subsequent accuracy improvements, weakening the practical relevance of the results.

---

> ### Author Rebuttal · Authors · 2025-07-31
>
> **Note:** We thank the reviewer for their very thoughtful feedback and comments. Following the reviewers’ suggestions, we conducted multiple experiments to strengthen our work. Since there is no dedicated common space for experiments, **we present them at the beginning of the responses to reviewers 8fbK (part 1) and J3ay (part 2)**.
>
> ---
> ---
>
> ## Response to WmN5
>
> > **Reviewer:** The experiment conducted by the author is quite small in scale, as CIFAR-10, CIFAR-100, and mini-ImageNet each contain only 50,000 images. The results could be easily influenced by overfitting, especially in the presence of noisy samples. I am curious whether this phenomenon still holds when the number of samples per class increases or when the data becomes noisier. Nowadays, pretraining typically requires massive and often noisy datasets. Further experiments should be conducted to verify the author’s claims.
>
> Thanks for raising this point. We believe that the reported results are not due to overfitting:
>
> 1. We conducted a wide range of experiments across multiple datasets, and as shown in Tab. 1 and Fig.  4, DCL‑trained models transfer well to downstream tasks with simple adaptation methods (e.g., LP, NCCC).
>
> 2. Beyond CIFAR and mini‑ImageNet, we also used SVHN, which contains 600 k images, and observed the same trends (Fig. 1).
>
> 3. Following the reviewer’s suggestion to scale up, we added experiments on Tiny‑ImageNet (200 classes) with ViT backbone (see experiment (2) in the response to 8fbK) and ImageNet‑1k (1000 classes) with ResNet50 backbone (see experiment (1) in the response to 8fbK). These datasets not only have a significantly larger number of classes but also naturally contain more label noise and sample‑level ambiguity than CIFAR‑like datasets.
>
> We would also be happy to explore specific experiments with noisier data if the reviewer has a concrete suggestion.
>
> > **Reviewer:** The central claim that standard contrastive learning approximates a supervised variant under a growing number of classes is conceptually intuitive and has been suggested in prior works. While Theorem 1 formalizes this intuition with a clean bound, the assumption that the number of classes tends to infinity makes the result less surprising and arguably less practical. The paper ultimately confirms that contrastive objectives yield clustering behavior similar to supervised objectives, but this is already an implicit assumption in much of the literature. Thus, the theoretical contribution, though correct, feels incremental rather than insightful.
>
> We appreciate the reviewer’s perspective and the opportunity to clarify the novelty and practical relevance of our results.
>
> 1. Incremental contribution: We agree that it is intuitive to view self-supervised learning (SSL) and supervised learning (SL) as closely related, and we acknowledge that the literature often assumes clustering properties. However, these are very strong and largely unsubstantiated assumptions. **Our work provides concrete theoretical guarantees that explicitly establish the relationship between SSL and SL and explain not only why clustering arises, but also what kind of clustering is expected.**
>
> Thm. 1 provides a label‑agnostic bound that directly relates CL and SL. It informs us what kind of supervised loss is most related to CL (e.g., NSCL), it holds for all possible labelings simultaneously and does not rely on data‑distributional, augmentation, or architectural assumptions, is substantially more general than previous heuristic arguments (see lines 80–91) that depend on strong, often unrealistic conditions. In response to reviewer feedback, we further substantiated this result by showing in (experiment (7) in the response to J3ay) that the learned representations from DCL and NSCL are highly aligned.
>
> Prop. 1 and its associated experiments make an equally important contribution: they (i) characterize the specific clustering properties that hold in practice for CL‑trained models, and (ii) demonstrate why these properties benefit downstream tasks. Since clustering is often *assumed* rather than analyzed in the literature, these results fill a crucial gap in our understanding of SSL.
>
> 2. Beyond the $C \to \infty$ regime: While the gap in Thm. 1 tightens as the number of classes increases, the theorem does not require $C \to \infty$. As shown in Figs. 2-3, the bound is already tight and correlates strongly with the observed loss gap for moderate class counts (e.g., 10–100 classes in CIFAR‑10/100 and mini‑ImageNet).
>
> In summary, our contributions (i) formalize a widely cited but previously speculated intuition, (ii) validate it in realistic settings, and (iii) link it directly to downstream performance. We therefore believe the work goes well beyond incremental progress, offering genuinely new theoretical and empirical insights into why SSL methods succeed.
>
> > **Reviewer:** This paper does not provide practical guidance for designing CL systems. For example, it does not address important questions such as how augmentation affects learning, the robustness of the theoretical results under incomplete augmentation, or whether CL can outperform supervised learning in all cases. Although the authors introduce an upper bound on the few-shot error, it relies on quantities that are difficult to explicitly control during learning (e.g., class-wise variance) and has limited practical impact on accuracy. Moreover, minimizing loss does not directly translate into subsequent accuracy improvements, weakening the practical relevance of the results.
>
> We thank the reviewer for raising important questions about the practical implications of our work. While our primary focus is not on proposing new practical training techniques, our work provides a foundational framework for explaining why existing self-supervised methods, such as CL and DCL, are theoretically sound, what aspects make them so successful, and what kinds of representations they capture. While these insights may not directly tell practitioners how to choose training hyperparameters, they offer a more informed way to analyze and ultimately improve CL systems in practice.
>
> * **Relationship with supervised learning:** There is a growing intuition that self-supervised learning and supervised learning are closely related, but it is only through the bound in Theorem 1 and the discussion in lines 155–161 that we identify the mechanism behind this relationship. Our bound quantifies the gap between SSL and supervised learning and clarifies which supervised method (e.g., NSCL) is most closely related to CL/DCL training. In fact, our new empirical results (experiment (7) in the response to J3ay) verify that by showing a very strong alignment between the learned embeddings of CL/DCL and NSCL.
>
> * **Practical impact of the few-shot error bound:** The reviewer is correct that our bound depends on quantities not directly controlled during training and that it is not a direct estimator of accuracy. However, we argue this is a feature, not a limitation, because its purpose is to explain rather than serve as a new training objective.
>
> A useful analogy is the bias–variance tradeoff. The formula decomposing error into bias, variance, and irreducible error is one of the most important concepts for practitioners. Although one cannot directly control bias or variance during a single training run, the decomposition provides a powerful framework for diagnosing failures (e.g., high variance implies overfitting). This understanding then guides practical decisions such as increasing regularization or gathering more data.
>
> Similarly, our bound in Prop. 1 is an analytical tool for understanding contrastive learning. Its primary value lies not in the final number it produces but in how it decomposes error and introduces new vocabulary for analyzing performance. Specifically, it distinguishes between two different quantities: within-class variance (CDNV) and the variance along the directions separating class centers (**directional CDNV**). This analytical lens leads to a core finding presented in Fig. 5: successful CL training should significantly reduce directional CDNV while maintaining a relatively stable CDNV. This is a non-obvious insight that explains how label-agnostic methods learn linearly separable features.

---

> > ### Comment · Reviewer_WmN5 · 2025-08-05
> >
> > Thank the authors for the detailed and thoughtful rebuttal. I appreciate the additional experiments on larger and noisier datasets such as Tiny-ImageNet and ImageNet-1k, which help address concerns regarding scalability and robustness. These results strengthen the empirical support for the paper’s claims. The rebuttal also clarifies the theoretical contributions, highlighting the novelty and generality of the provided bounds and their connections to both supervised and self-supervised paradigms.
> >
> > While some limitations remain in terms of direct practical guidance, the paper provides a solid theoretical foundation that could inform future design and analysis of contrastive learning systems. I will raise my score.

---

> > > ### Author Response · Authors · 2025-08-06
> > >
> > > Thank you so much for your constructive feedback and for reviewing our response. We’re glad to hear that the additional experiments helped address your concerns, and we will be sure to include them in the final version. We also appreciate your recognition of the theoretical contributions, and we’re happy that our clarifications highlighted the novelty and generality of our results, their relevance to both supervised and self-supervised paradigms, and the importance of our work as a tool for the design and analysis of contrastive learning methods.
> > >
> > > Thank you again for taking the time to engage with our work. We are truly grateful for your thoughtful comments and for your decision to raise your score.

---

> ### Comment · Area_Chair_9oLT · 2025-08-05
> **post-rebuttal comments**
>
> Dear reviewer WmN5,
>
> As you may have seen already, the authors have responded to the questions in your initial review. Can you please share your thoughts post rebuttal, once you've had a chance to see the author responses and also the other reviews?
>
> This is critical to arrive at an informed decision.
>
> Best, AC

---

### Official Review · Reviewer_8fbK · 2025-07-03

**Clarity:** 4
**Significance:** 3
**Originality:** 3
**Rating:** 5
**Confidence:** 3

**Summary:**

This paper provides a theoretical and empirical bridge between self-supervised contrastive learning (CL) and a supervised variant called negatives-only supervised contrastive learning (NSCL). The authors show that as the number of classes increases, CL implicitly approximates NSCL, even without label supervision. They prove that NSCL leads to representations exhibiting neural collapse structure, and they propose a new bound for few-shot linear probing error based on feature variability. Experiments on image classification benchmarks support the theoretical claims, showing that CL-trained models, despite lacking labels, learn semantically meaningful and transferable representations.

**Questions:**

1. The experiments show that the gap between DCL and NSCL decreases as the number of classes increases, which aligns well with the theoretical analysis. However, the performance difference between the two methods remains noticeable. More discussion on this observation could help clarify how much of the theoretical alignment translates into practical outcomes.
2. In addition to DCL and NSCL, it would be helpful to include comparisons with standard CL methods. Based on the paper’s analysis, the gap between CL and NSCL is expected to be even larger, which could further support the theoretical claims.
3. Can the authors quantify the similarity of the learned representations—not just the loss values—between CL/DCL and NSCL-trained models? For example, using clustering metrics and others.

**Ethical Concerns:**

["NO or VERY MINOR ethics concerns only"]

**Final Justification:**

The authors have provided a rebuttal, including extensive new experiments across multiple datasets and architectures. These results support the theoretical claims and address the concerns raised in my initial review. I also agree with reviewer 8nSm that clearly distinguishing the roles of CL and DCL in the revised version would help improve the clarity and impact of the paper.

**Limitations:**

While the paper presents clear analysis and supporting experiments, further evaluation on a wider range of models and larger-scale datasets would be helpful. This could clarify how well the findings generalize beyond the current experimental setting.

**Paper Formatting Concerns:**

.

**Quality:**

4

**Strengths And Weaknesses:**

The paper presents a compelling and well-founded theoretical connection between unsupervised and supervised contrastive losses. The experimental results consistently support the theoretical claims.
However, despite the clear structure and explanations, several questions remain—such as the performance gap between DCL and NSCL, comparisons with standard CL, and the degree of equivalence between the learned representations.

---

> ### Author Rebuttal · Authors · 2025-07-31
>
> **Note:** We thank the reviewer for their very thoughtful feedback and comments. Following the reviewers’ suggestions, we conducted multiple experiments to strengthen our work. Since there is no dedicated common space for experiments, **we present them at the beginning of the responses to reviewers 8fbK (part 1) and J3ay (part 2)**.
>
> ---
> ---
>
> ## Summary of experiments (part 1)
>
>
> **Experiment (1)** We reproduced Fig. 2 (top) with SimCLR-v2 with the ResNet-50 architecture for **ImageNet-1k (1000 classes)**.
>
> | **Epochs** | **CL** | **DCL** | **NSCL** | **Upper Bound** |
> | ---------- | ------ | ------- | -------- | --------------- |
> | **0** | 6.9156 | 6.9146 | 6.9136 | 6.9210 |
> | **10** | 6.1298 | 6.1276 | 6.1265 | 6.1338 |
> | **50** | 6.0995 | 6.0972 | 6.0960 | 6.1034 |
> | **100** | 6.1165 | 6.1142 | 6.1130 | 6.1203 |
>
> **Experiment (2)** We reproduced the experiments in Fig. 2 (top) for SimCLR-v2 with the **ViT_base architecture**. The experiments were conducted for **TinyImageNet (200 classes)**, CIFAR10, CIFAR100, and mini-ImageNet.
>
> ### *CIFAR10*
>
> | **Epochs** | **CL**   | **DCL**  | **NSCL** | **Upper Bound** |
> |------------|----------|----------|----------|-----------------|
> | **0**      | 7.4213   | 7.4206   | 7.3130   | 7.9125          |
> | **10**     | 6.8764   | 6.8753   | 6.7488   | 7.3483          |
> | **100**    | 6.7430   | 6.7418   | 6.6040   | 7.2034          |
> | **1000**   | 6.7128   | 6.7114   | 6.5706   | 7.1699          |
>
> ### *CIFAR100*
>
> | **Epochs** | **CL**   | **DCL**  | **NSCL** | **Upper Bound** |
> |------------|----------|----------|----------|-----------------|
> | **0**      | 7.3633   | 7.3627   | 7.3522   | 7.4241          |
> | **10**     | 6.9615   | 6.9605   | 6.9494   | 7.0213          |
> | **100**    | 6.8088   | 6.8077   | 6.7964   | 6.8684          |
> | **1000**   | 6.7659   | 6.7648   | 6.7534   | 6.8254          |
>
> ### *Mini-imagenet*
>
> | **Epochs** | **CL**   | **DCL**  | **NSCL** | **Upper Bound** |
> |------------|----------|----------|----------|-----------------|
> | **0**      | 7.1455   | 7.1447   | 7.1345   | 7.2065          |
> | **10**     | 6.6273   | 6.6259   | 6.6145   | 6.6865          |
> | **100**    | 6.5006   | 6.4990   | 6.4875   | 6.5595          |
> | **1000**   | 6.3979   | 6.3962   | 6.3839   | 6.4559          |
>
> ### *Tiny-imagenet (C=200)*
>
> | **Epochs** | **CL**   | **DCL**  | **NSCL** | **Upper Bound** |
> |------------|----------|----------|----------|-----------------|
> | **0**      | 6.7862   | 6.7851   | 6.7800   | 6.8165          |
> | **10**     | 6.2234   | 6.2213   | 6.2158   | 6.2523          |
> | **100**    | 6.0748   | 6.0725   | 6.0669   | 6.1034          |
> | **500**    | 6.0342   | 6.0317   | 6.0261   | 6.0625          |
>
> **Experiment (3)** We reproduced the experiment in Fig. 4 (bottom) with a pre-trained ResNet-50 that was **trained on ImageNet-1k** and **evaluated the model on both ImageNet-1k and TinyImageNet**.
>
> ### *mini-imagenet*
>
> |      Num-shots >         | **1**    | **5**    | **10**   | **20**   | **50**   | **100**  | **200**  | **500**  |
> |---------------|----------|----------|----------|----------|----------|----------|----------|----------|
> | **err_nccc**  | 0.2446 | 0.0866 | 0.0593 | 0.0526 | 0.0506 | 0.0413 | 0.0400 | 0.0400 |
> | **err_bound** | 55.3146 | 9.4475 | 5.3099 | 3.2047 | 1.7806 | 1.2012 | 0.8969 | 0.6788 |
>
>
> ### *Full-imagenet*
>
> |      Num-shots >         | **1**    | **5**    | **10**   | **20**   | **50**   | **100**  | **200**  | **500**  |
> |---------------|----------|----------|----------|----------|----------|----------|----------|----------|
> | **err_nccc**  | 0.2813 | 0.1000 | 0.0566 | 0.0420 | 0.0400 | 0.0340 | 0.0360 | 0.0340 |
> | **err_bound** | 56.9036 | 9.5940 | 5.3628 | 3.2189 | 1.7749 | 1.1896 | 0.8687 | 0.6502 |
>
>
> **Experiment (4)** We reproduced the experiment in Fig. 4 (bottom) with the **ViT_base architecture** for **CIFAR10, CIFAR100, mini-ImageNet**.
>
> ### *CIFAR10*
>
> |       Num-shots >       | **1**    | **5**    | **10**   | **20**   | **50**   | **100**  | **300**  | **1000** |
> |---------------|----------|----------|----------|----------|----------|----------|----------|----------|
> | **err_nccc**  | 0.3306   | 0.1357   | 0.0713   | 0.0360   | 0.0195   | 0.0176   | 0.0159   | 0.0153   |
> | **err_lp**    | 0.1661   | 0.0727   | 0.0357   | 0.0463   | 0.0347   | 0.0266   | 0.0160   | 0.0058   |
> | **err_bound** | 37.2999  | 6.1490   | 3.4042   | 2.0238   | 1.1013   | 0.7299   | 0.4423   | 0.3141   |
>
> ### *CIFAR100*
>
> |      Num-shots >        | **1**    | **5**    | **10**   | **20**   | **50**   | **100**  | **200**  | **500**  |
> |---------------|----------|----------|----------|----------|----------|----------|----------|----------|
> | **err_nccc**  | 0.2973   | 0.1457   | 0.1052   | 0.0782   | 0.0622   | 0.0583   | 0.0537   | 0.0509   |
> | **err_lp**    | 0.2239   | 0.0970   | 0.0749   | 0.0599   | 0.0509   | 0.0420   | 0.0350   | 0.0290   |
> | **err_bound** | 32.6582  | 5.5778   | 3.1349   | 1.8920   | 1.0512   | 0.7091   | 0.5295   | 0.4007   |
>
> ### *Mini-imagenet*
>
> |      Num-shots >         | **1**    | **5**    | **10**   | **20**   | **50**   | **100**  | **200**  | **500**  |
> |---------------|----------|----------|----------|----------|----------|----------|----------|----------|
> | **err_nccc**  | 0.2210   | 0.0922   | 0.0679   | 0.0651   | 0.0530   | 0.0517   | 0.0488   | 0.0460   |
> | **err_lp**    | 0.2500   | 0.0820   | 0.0540   | 0.0459   | 0.0450   | 0.0500   | 0.0379   | 0.0240   |
> | **err_bound** | 65.0692  | 10.0063  | 5.3739   | 3.0990   | 1.6178   | 1.0365   | 0.6889   | 0.4433   |
>
>
> ---
> ---
>
> ## Response to 8fbK
>
> > **Reviewer:** The experiments show that the gap between DCL and NSCL decreases as the number of classes increases, which aligns well with the theoretical analysis. However, the performance difference between the two methods remains noticeable. More discussion on this observation could help clarify how much of the theoretical alignment translates into practical outcomes.
>
> The loss‑gap bound in Thm. 1 is label‑agnostic: it shows how DCL incentivizes the embeddings to support any labeling (see lines 155–161), but it does not imply that the learned embedding geometry will match a specific labeling, as NSCL does. In fact, under the unconstrained features model and assuming $d > N$, any global minimizer of DCL forms an $N$‑way equiangular tight frame (ETF). This configuration drives directional‑CDNV to 0 even when the full CDNV remains non‑zero.
>
> As observed in Figs. 5, DCL effectively drives the directional‑CDNV to low values, while Thm. 2 for NSCL implies that both directional and regular CDNVs converge to low values. As predicted by Prop. 1 (and verified in Fig. 4 (bottom)), this difference explains why DCL achieves reasonable downstream performance, but NSCL—by driving both CDNV measures lower—ultimately performs even better.
>
> In conclusion, we do not expect DCL and NSCL to learn the exact same representations. However, our theory characterizes their relationship, clarifies the types of representations produced by DCL training, and explains why these representations remain effective for downstream tasks using linear probes.
>
> > **Reviewer:** In addition to DCL and NSCL, it would be helpful to include comparisons with standard CL methods. Based on the paper’s analysis, the gap between CL and NSCL is expected to be even larger, which could further support the theoretical claims.
>
> Following the reviews, we conducted an experiment to reproduce Fig. 2 (top) for CL-training (see experiment (6) in the response to J3ay). In addition, we evaluated CL, DCL and NSCL and the bound for DCL/CL-trained models and observed negligible differences in the values of CL and DCL (see experiments (1), (2) and (6) in the responses to 8fbK and J3ay).
>
> Theoretically, CL and DCL differ only by removing the anchor sample from the denominator, and the proof of Thm. 1 applies to both. Specifically:
> $$
> \mathcal L_{\text{NSCL}} \le \mathcal L_{\text{DCL}} \le \mathcal L_{\text{CL}}
> \quad\text{and}\quad
> \mathcal L_{\text{CL}}-\mathcal L_{\text{NSCL}}
> \le \log\Bigl(1+\tfrac{e^{2}}{C-1}\Bigr).
> $$
> This implies
> $$
> 0 \le \mathcal L_{\text{DCL}}-\mathcal L_{\text{NSCL}}
> \le \mathcal L_{\text{CL}}-\mathcal L_{\text{NSCL}}
> \le \log\Bigl(1+\tfrac{e^{2}}{C-1}\Bigr).
> $$
>
> > **Reviewer:** Can the authors quantify the similarity of the learned representations—not just the loss values—between CL/DCL and NSCL-trained models? For example, using clustering metrics and others.
>
> We conducted new experiments to quantify the representation similarity between CL/DCL- and NSCL-trained ViT models (see experiment (7) in the response to J3ay).
>
> > **Reviewer:** While the paper presents clear analysis and supporting experiments, further evaluation on a wider range of models and larger-scale datasets would be helpful. This could clarify how well the findings generalize beyond the current experimental setting.
>
> Thank you for the suggestion. Following the reviews, we conducted extensive experiments with the ViT architecture, as well as the ImageNet-1k (1000 classes) and Tiny-ImageNet (200 classes) datasets (see experiments (1)-(5) in the responses to 8fbK and J3ay).

---

> > ### Comment · Reviewer_8fbK · 2025-08-05
> >
> > Thank you for the detailed response. The rebuttal provides extensive additional experiments that further strengthen the paper’s claims and address my concerns. I hope these results will be clearly included in the revised version. In line with reviewer 8nSm’s comments, I also encourage the authors to clarify the usage of CL and DCL, as doing so would improve readability and help avoid potential misunderstandings. My overall assessment remains positive, and I will keep my score unchanged.

---

> > > ### Author Response · Authors · 2025-08-06
> > >
> > > Thank you for your continued support and thoughtful feedback. We’re glad that our additional experiments and clarifications addressed your concerns. As suggested, we will revise the text to clarify the use of CL and DCL to improve clarity and prevent potential misunderstandings. We will also ensure that all new experiments are clearly included in the final version.
> > >
> > > Thank you once again for your appreciation of our work and constructive comments.

---

> ### Comment · Area_Chair_9oLT · 2025-08-05
> **post-rebuttal comments**
>
> Dear reviewer 8fbK,
>
> As you may have seen already, the authors have responded to the questions in your initial review. Can you please share your thoughts post rebuttal, once you've had a chance to see the author responses and also the other reviews?
>
> Best, AC

---

### Decision · Program_Chairs · 2025-09-17

**Decision:**

Accept (poster)

**Comment:**

This paper tackles the issue of insufficient theoretical foundations for self-supervised contrastive learning, in particular decoupled contrastive learning (DCL). This is done by approximating the DCL loss function with NSCL (negatives-only supervised contrastive loss). This approximation is shown to be justified when the number of classes increases. Given this, the geometry of the representation learned by NSCL is studied to demonstrate properties such as augmentation collapse, within-class collapse, etc. The paper then shows convincing empirical evaluation.

Several positives were identified during the review process. For example, the paper provides a new perspective on the theoretical foundations for self-supervised contrastive learning. This was noted by all the reviewers and the AC. Another important aspect is the strong empirical evaluation on multiple datasets.

After the reviewer-author and the reviewers-AC discussion phases, two questions remain regarding this paper: (1) the need to use DCL instead of CL for better clarity, and (2) the fact that the paper analyzes "the geometry of the representation learned by NSCL instead of (D)CL", as noted by reviewer 8nSm. The first point can be addressed relatively easily by the authors (which they agreed to take note of), and while the second point is indeed very relevant, the paper has sufficient merit in presenting a theoretical analysis of self-supervised (decoupled) contrastive learning. Given the overall positive comments from all the reviewers, the recommendation is to accept this paper. The final version should include all the clarifications provided in the rebuttal and the discussions that followed.